# SHAKTI: Subglacial Hydrology And Kinetic, Transient Interactions v1.0

Aleah Sommers[1], Harihar Rajaram[1], and Mathieu Morlighem[2]

[1]Department of Civil, Environmental, and Architectural Engineering, University of Colorado, Boulder, Colorado, USA
[2]Department of Earth System Science, University of California, Irvine, California, USA

*Correspondence to:* Aleah Sommers (aleah.sommers@colorado.edu)

**Abstract.** Subglacial hydrology has a strong influence on glacier and ice sheet dynamics, particularly through the dependence of sliding velocity on subglacial water pressure. Significant challenges are involved in modeling subglacial hydrology, as the drainage geometry and flow mechanics are constantly changing, with complex feedbacks that play out between water and ice. A clear tradition has been established in the subglacial hydrology modeling literature of distinguishing between channelized (efficient) and sheetlike (inefficient or distributed) drainage systems or components and using slightly different forms of the governing equations in each subsystem to represent the dominant physics. Specifically, many previous subglacial hydrology models disregard opening by melt in the sheetlike system or redistributing it to adjacent channel elements in order to avoid runaway growth that occurs when it is included in the sheetlike system. We present a new subglacial hydrology model, SHAKTI (Subglacial Hydrology And Kinetic, Transient Interactions), in which a single set of governing equations is used everywhere, including opening by melt in the entire domain. SHAKTI employs a generalized relationship between the subglacial water flux and the hydraulic gradient that allows representation of laminar, turbulent, and transitional regimes, depending on the local Reynolds number. This formulation allows for coexistence of these flow regimes in different regions, and the configuration and geometry of the subglacial system evolves naturally to represent sheetlike drainage as well as systematic channelized drainage under appropriate conditions. We present steady and transient example simulations to illustrate the features and capabilities of the model, and to examine sensitivity to mesh size and time-step size. The model is implemented as part of the Ice Sheet System Model (ISSM).

## 1 Introduction

One of the significant consequences of contemporary climate change is rising sea level. A large component of sea level rise is due to the transfer of ice from glaciers and ice sheets into the ocean via melt, runoff, and iceberg calving (Church et al., 2013). Future ice dynamics remains a major uncertainty in sea level rise predictions involving many uncertain factors, including basal lubrication and effects on sliding velocities from subglacial drainage (e.g., Church et al., 2013; Shannon et al., 2013).

Although massive outlet glaciers of West Antarctica may be on the verge of irreversible collapse in the next 200 to 1,000 years (Joughin et al., 2014; DeConto and Pollard, 2016), the Greenland ice sheet is currently the single largest contributor to sea level rise (Shepherd et al., 2012). Considering the substantial amount of water held in this frozen reservoir, it is important to

improve understanding of its behavior, including the subtleties of its drainage, which affects ice velocity through sliding. Since 1990, many Greenland outlet glaciers have displayed dramatic accelerations and frontal retreats, yielding substantial changes on the rapid timescale of decades or years (Joughin et al. 2010). Other glaciers, however, have accelerated less rapidly or even decelerated over the same period (McFadden et al., 2011), and the mechanisms driving these contrasting responses are still not

entirely understood. The recent accelerations observed in marine terminating outlet glaciers, which exhibit some of the greatest accelerations and are highly sensitive to changes in terminus conditions, may be in response to changing ocean temperatures (Nick et al., 2009, Rignot et al., 2010, Andresen et al., 2012), but their diverse behaviors have been found to depend on more factors than ocean temperature alone, such as bed topography and subglacial discharge distribution (Slater et al., 2015; Rignot et al., 2016). In land terminating glaciers, the observed accelerations are likely driven largely by water inputs to the ice sheet

from the surface via crevasses and moulins, similar to alpine glaciers (e.g., Anderson et al., 2004; Bartholomaus et al., 2008). Meltwater inputs have been shown to drive variation in ice velocities on the Greenland ice sheet (e.g., Zwally et al., 2002; Bartholomew et al., 2012), as well as seasonal changes in the efficiency of the subglacial drainage system (e.g., Bartholomew et al., 2010; Chandler et al., 2013; Cowton et al., 2013; Andrews et al., 2014).

The hydrology of meltwater on the surface, within, and beneath glaciers and ice sheets should ideally be viewed and modeled

as a complex system of processes, considering the interconnectedness of surface mass balance, meltwater retention, discharge at the ice margin, and feedbacks between hydrology and ice dynamics (e.g., Rennermalm et al., 2013; Nienow et al., 2017). Water delivered to the bed through englacial conduits drives basal sliding, which has important effects on flow in some regions (Vaughan et al., 2013), and year-round sliding can occur with temperate bed conditions (Colgan et al., 2011). Increased melt-water input to the bed, however, does not necessarily imply increased basal sliding, contrary to what might seem intuitive. For

example, as meltwater input increases, water pressure under the ice increases, leading to enhanced basal lubrication and higher sliding velocity (Zwally et al., 2002). But with sustained meltwater input over a melt season, more efficient drainage channels can develop, decreasing the water pressure (Schoof, 2010). Characteristics of individual outlet glaciers such as bed topography, ice geometry, surface temperature, and other factors all play into the intricate choreography of the seasonal evolution of the subglacial drainage system and its influence on ice velocity. Subglacial hydrology models have had success in simulating

realistic drainage behavior, but challenges still remain.

The goal of this modeling effort is to see if we could use a single set of governing equations to produce systematic, self-organized channelization where it should occur. In this paper, we describe the model formulation of SHAKTI (Subglacial Hydrology And Kinetic, Transient Interactions), which allows for flexible evolution of the subglacial drainage system configuration and flow regimes using a single set of governing equations over the entire domain. The model aims to represent the

complex interactions due to (kinetic) movement of ice and water and (transient) changes of the subglacial system through time. We hope this unified formulation may be used to facilitate exploration of the conditions under which different drainage system types form and persist, and the flow regimes experienced in different areas of a domain. With upcoming application to actual glaciers, this type of model could provide useful insights into the seasonal evolution of real subglacial drainage systems and their influence on mass loss from the Greenland ice sheet, with the potential for broader application to Antarctica and alpine

glaciers.

The paper is structured as follows: in Sections 1.1-1.2, we provide a brief summary and review of historical and recent subglacial hydrology modeling progress to put our model in context. We then present the model's governing equations and the numerical framework in Section 2, with illustrative simulations to demonstrate key model features and capabilities in Section 3, and a discussion of implications and model limitations in Section 4.

## 1.1 Subglacial hydrology modeling context

Subglacial hydrology has long been an area of interest, initially in the context of geomorphology, groundwater, and surface hydrology from alpine glaciers, and more recently in the context of its influence on ice sheet dynamics. Below is a brief and selective summary of previous subglacial hydrology modeling work motivated by glacier sliding. We direct readers to Flowers (2015) for a comprehensive review of the full subject history, recent advancements, and current challenges.

The first major efforts to quantitatively model subglacial hydrology began in the 1970s. Shreve (1972) described a system of arborescent subglacial channels, and Röthlisberger (1972) formulated equations for semi-circular channels melted into the base of the ice sheet, in a state of equilibrium between melt opening and creep closure. Nye (1973) expanded the work of Röthlisberger to consider channels incised into bedrock or subglacial sediments, and more fully developed the equations into models for explaining outburst floods (Nye, 1976). In a different approach, Weertman (1972) considered subglacial drainage through a water sheet of approximately uniform thickness. In the following decade, different plausible drainage configurations were also proposed, such as a system of "linked cavities", spaces that open behind bedrock bumps as a result of glacier sliding (Walder, 1986; Kamb, 1987). By the mid-1980s, it was recognized that the major components of subglacial hydrology could be classified as either efficient (channels or canals) or inefficient (thin sheets, flow through porous till, or distributed systems of linked cavities, often represented in continuum models as a sheet). While channels themselves emerge as a result of self-organized selective growth from a linked cavity system, a clear distinction between these two subsystems was established.

Since 2000, a renewed surge of interest in subglacial hydrology has been sparked as mass loss increases from glaciers and ice sheets and sea level rise is increasingly perceived as an imminent reality, generating a flurry of new observations and modeling advances. Although the effects of surface melt on ice sheet dynamics are not yet entirely understood (e.g., Clarke, 2005; Joughin et al., 2008), observations have reinforced the fact that surface meltwater significantly influences flow behavior in alpine glaciers and ice sheets (e.g., Mair et al., 2002; Zwally et al., 2002; Bartholomaus et al., 2008; Howat et al., 2008; Shepherd et al., 2009; Bartholomew et al., 2010; Hoffman et al., 2011; Sundal et al., 2011; Bartholomew et al., 2012; Meierbachtol et al., 2013; Andrews et al., 2014). Along with more detailed observations, several efforts were made in the early 2000s to accurately simulate subglacial hydrology. Some of these studies treated the subglacial system as a water sheet of uniform thickness (e.g., Flowers and Clarke, 2002; Johnson and Fastook, 2002; Creyts and Schoof, 2009; LeBrocq et al., 2009). Arnold and Sharp (2002) presented a model with both distributed and channel flow, but only one configuration could operate at a time. Kessler and Anderson (2004) introduced a model using discrete drainage pathways that could transition between distributed and channelized modes, and Flowers et al. (2004) used a combination of a distributed sheet in parallel with a network of efficient channels. Schoof (2010) developed a 2D network of discrete conduits that could behave like either channels or cavities, and found that with sufficiently large discharge an arborescent network of channel-like conduits would

form, although the resulting geometry was highly dependent on the rectangular grid used. Hewitt (2011) developed a model that used a water sheet to represent evolving linked cavities averaged over a patch of bed (an effective porous medium), coupled to a single channel.

More recent studies tied together key elements of subglacial drainage to form increasingly realistic 2D models. Hewitt (2013) introduced a linked-cavity continuum sheet integrated with a structured channel network. In that model, channels open by melt, while the distributed sheet opens only by sliding over bedrock bumps (neglecting opening by melt from dissipative heat). Melt from dissipative heat contributes only to opening in channels. Werder et al. (2013) presented a model that involves water flow through a sheet (representative of averaged linked cavities) along with channels that are free to form anywhere along edges of the unstructured numerical mesh, exchanging water with the surrounding distributed sheet. Approaching the problem in a different way, Bougamont et al. (2014) reproduced seasonal ice flow variability through the hydro-mechanical response of soft basal sediment in lieu of simulating the evolution of a subglacial drainage system. To capture broad characteristics of subglacial drainage without resolving individual elements, DeFleurian et al. (2014) employed a 2D dual-layer porous medium model, and Bueler and Pelt (2015) formulated equations for a 2D model that combines water stored in subglacial till with linked cavities. To help explain observations of high water pressure in late summer and fall, recent observations and modeling efforts have highlighted the importance of representing hydraulically isolated or "weakly connected" regions of the bed (Hoffman et al., 2016; Rada and Schoof, 2018), and addressed the problem by facilitating seasonal changes in the hydraulic conductivity (Downs et al., 2018).

## 1.2 Distinction between channelized and sheetlike drainage, and the problem of dissipation

A common theme in the subglacial hydrology modeling literature is a distinction between channelized (efficient) and sheetlike (inefficient or distributed) drainage systems or components. In most existing 2D models, either only one of these forms is considered, or else slightly different equations are applied to coupled channel and sheet components. For the sheetlike system, these models only consider opening (i.e. growth of the sheet thickness) due to sliding over bedrock bumps, disregarding opening by melting of the upper ice surface. Melt is generated by the thermal energy obtained from dissipated mechanical energy (commonly referred to as energy loss or head loss). However, these models redirect the generated thermal energy into adjacent channel components that are allowed to melt and grow. Channel components are allowed to form in pre-specified locations or to evolve along the edges of sheetlike elements, as in Werder at el. (2013). The main reason that most of these models disregard melt opening in the sheetlike system is to avoid the unstable behavior that has been found to occur when it is included, leading to unstable growth where the melt opening rate exceeds the closure rate, sparking channelization (Hewitt, 2011) or driving initiation of glacial floods (Schoof, 2010). The transition to a channelized state has been described elegantly in previous work (e.g., Walder, 1986; Kamb, 1987; Schoof, 2010; Hewitt, 2011; Schoof et al., 2012; Werder et al., 2013; Hoffman and Price, 2014).

In reality, the subglacial hydrologic system is comprised of a wide array of drainage features, of which the sheet and channel are two end-members. Imposing a sharp distinction between the treatment of the melt opening term, and dividing the governing equations between different model components may not allow for the full array of drainage features to arise. It is also

a bit artificial to redirect the opening by melt in sheetlike elements to nearby channels. In the model formulation described in this paper, a single set of governing equations is applied over the entire domain, including the melt opening term everywhere. In our formulation, the hydraulic transmissivity of the subglacial domain is allowed to vary spatially and temporally, allowing for a continuum of drainage features. We also account for laminar, turbulent and intermediate flow regimes, based on an experimentally verified flow law for rough-walled rock fractures (Zimmerman et al., 2004). The gap thickness of each computational element in a discretization of the governing equations is allowed to evolve flexibly, and sequential elements with high gap growth rates typically link up to produce channelized features. The ability to represent co-existing turbulent, laminar and intermediate regimes regimes appears to be a promising approach to overcoming the previously mentioned instability that occurs when the melt generated by mechanical energy dissipation is retained in the sheet system equations. Even with the melt opening term included everywhere in the domain, we are able to generate steady and transient drainage configurations that include channel-like efficient drainage pathways. Our model does not aim to simulate every individual cavity or specific channel cross-section, but rather captures the homogenized effects of these elements on a discrete mesh. As we demonstrate in Section 3, although the resolution of subglacial geometry in our approach is mesh/grid-sensitive, the patterns of simulated basal water pressure and effective pressure (which are most relevant for calculating sliding velocities in ice dynamics models) are relatively robust with coarse resolutions ($\sim 400$ m).

## 2 SHAKTI model description

This flexible subglacial hydrology model can handle transient meltwater inputs, both spatially distributed and localized, and allows the basal water flux and geometry to evolve according to these inputs to produce flow and drainage regimes across the spectrum from sheetlike to channelized. The subglacial drainage system is represented as a sheet with variable gap height, and we employ a flux formulation based on fracture flow equations. Channelized locations are not prescribed a priori, but can arise and decay naturally as reflected in self-organized formation of connected paths of large gap height (calculated across elements) and lower water pressure (calculated at vertices) than their surroundings. In contrast, previous models allow for efficient channels to arise along element or grid edges and calculate a specific cross-sectional channel area (e.g., Schoof, 2010; Hewitt et al., 2013; Werder et al., 2013).

The parallelized, finite element SHAKTI model is currently implemented as part of the Ice Sheet System Model (ISSM; Larour et al., 2012; http://issm.jpl.nasa.gov), with full two-way coupling with the ice dynamics model planned for upcoming work. Below, we present the equations involved in the SHAKTI formulation. The governing equations are similar to those used in Werder et al. (2013), with some key differences that enable application of the same set of equations everywhere in the domain.

### 2.1 Summary of model equations

The SHAKTI model is based upon governing equations that describe conservation of water and ice mass, evolution of the gap height, water flux (approximate momentum equation for water velocity integrated over the gap height), and internal melt

generation (approximate energy equation for heat produced at the bed). All variables used in the equations are summarized in Table 1, with constants and parameters summarized in Table 2.

In general, a complete set of governing equations for subglacial hydrology models should include acceleration terms in the momentum equation, and advection and in-plane conduction terms should be included in the energy equation. The most general form of the conservation equations for subglacial hydrology would be a multi-dimensional extension of the equations described by Spring and Hutter (1981) and Clarke (2003), with augmentation to account for opening by sliding. Our model formulation and most existing subglacial hydrology models typically neglect the acceleration terms in the momentum equation and employ an approximate energy equation in which all dissipated mechanical energy is locally used to produce melt, and the equations presented here should be viewed as an approximation to the more general equations.

The water mass balance equation is written as:

$$\frac{\partial b}{\partial t} + \frac{\partial b_e}{\partial t} + \nabla \cdot \boldsymbol{q} = \frac{\dot{m}}{\rho_w} + i_{e \to b} \tag{1}$$

where $b$ is subglacial gap height, $b_e$ is the volume of water stored englacially per unit area of bed, $\boldsymbol{q}$ is basal water flux, $\dot{m}$ is basal melt rate, and $i_{e \to b}$ represents the input rate of surface meltwater from the englacial to subglacial system. This water balance assumes that the subglacial gap is always filled with water, and that water is incompressible.

Evolution of the gap height (subglacial geometry) involves opening due to melt and sliding over bumps on the bed, and closing due to ice creep:

$$\frac{\partial b}{\partial t} = \frac{\dot{m}}{\rho_i} + \beta u_b - A|p_i - p_w|^{n-1}(p_i - p_w)b \tag{2}$$

where $A$ is the ice flow law parameter, $n$ is the flow law exponent, $p_i$ is the overburden pressure of ice, $p_w$ is water pressure, $\beta$ is a dimensionless parameter governing opening by sliding, and $u_b$ is the magnitude of the sliding velocity. Equation (2) may be viewed as a generalized ice mass balance equation, augmented to consider opening by sliding. In most existing 2D models that include both channel and distributed sheetlike drainage components (e.g., Werder et al., 2013), melt opening is typically considered "channel opening" and opening by sliding over bumps on the bed is considered "cavity opening", with the different terms applied to the appropriate components within the model. Our model differs from other existing models in that we include both opening terms everywhere in the domain, similar to the conduit model of Schoof (2010). The opening by sliding parameter $\beta$ is a function of typical bed bump height ($b_r$) and bump spacing ($l_r$), as well as local gap height (so that opening by sliding only occurs where the gap height is less than the typical bump height). In defining $\beta$, we follow Werder et al. (2013):

$$\beta|_{b<b_r} = \frac{(b_r - b)}{l_r} \tag{3}$$

$$\beta|_{b \geq b_r} = 0 \tag{4}$$

The horizontal basal water flux (approximate momentum equation) is described based on equations developed for flow in rock fractures (e.g., Zimmerman et al. 2003, Rajaram et al. 2009, Chaudhuri et al. 2013):

$$\boldsymbol{q} = \frac{-b^3 g}{12\nu(1+\omega Re)} \nabla h \tag{5}$$

where $g$ is gravitational acceleration, $\nu$ is kinematic viscosity of water, $\omega$ is a dimensionless parameter controlling the nonlinear transition from laminar to turbulent flow, $Re$ is the Reynolds number, and $h$ is hydraulic head, defined as $h = p_w/(\rho_w g) + z_b$ (where $z_b$ is bed elevation). Note that the dimensions of the basal water flux are m$^2$ s$^{-1}$, i.e. a flow rate per unit width, obtained as an integral of the velocity profile across the gap thickness. The momentum Eq. (5) is approximate in the sense that acceleration terms are neglected and the flow is approximated as a locally plane shear flow. Equation (5) is a key piece of our model formulation, in that it allows for a spatially and temporally variable hydraulic transmissivity in the system, and facilitates representation of simultaneous coexistence of laminar, transitional and turbulent flow in subregions of the domain. Many existing subglacial hydrology models prescribe a hydraulic conductivity parameter and assume the flow to be turbulent everywhere. Equation (5) has been employed extensively for modeling flow in rock fractures, especially in the laminar flow regime ($\omega Re << 1$), wherein it is commonly referred to as the local cubic law. The extension of the local cubic law to transitional and turbulent flows, by incorporating a Reynolds number dependence as in Eq. (5), has also been employed in previous work on rock fractures (Zimmerman et al., 2004; Rajaram et al., 2009; Chaudhuri et al., 2013), and was experimentally verified by Zimmerman et al. (2009).

In the laminar flow regime, Eq. (5) derives from assuming locally plane Poiseuille flow and integrating the Stokes equations twice across the gap thickness to obtain:

$$\boldsymbol{q_{lam}} = \frac{-b^3 g}{12\nu} \nabla h \tag{6}$$

where $\nu$ is the kinematic viscosity of water. The definition of Reynolds number follows the precedent in fracture literature, using the gap height $b$ as a characteristic length scale:

$$Re = \frac{|v|b}{\nu} = \frac{|\boldsymbol{q}|}{\nu} \tag{7}$$

where $v$ is the average velocity across the gap . Note that for laminar flow, the flux in Eq. (6) is proportional to the hydraulic gradient $\nabla h$. The flux equation in the laminar regime (Eq. 6) is modified to allow for transition to a turbulent regime by introducing the additional term in the denominator to account for Reynolds number dependence. For fully developed turbulent flow with high Reynolds number ($\omega Re >> 1$ ), the magnitude of the flux $\boldsymbol{q}$ given by Eq. (5) is proportional to the square root of the magnitude of the hydraulic gradient:

$$|\boldsymbol{q_{turb}}|^2 = \frac{b^3 g}{12\omega} |\nabla h| \tag{8}$$

Equation (8) is analogous to the Darcy-Weisbach equation with a constant (i.e. not dependent on Reynolds number) friction factor for flow in ducts. For intermediate Reynolds numbers, Eq. (5) captures a nonlinear dependence between flux and hydraulic gradient that is in between the linear and square root dependences corresponding to laminar and turbulent flow regimes. The parameter $\omega$ controls the Reynolds number at which the deviation from the linear dependence becomes significant, and is also related to the friction factor. For example, with $\omega = 0.001$, $\omega Re$ is of order 10 at $Re = 10,000$, representing the value at which the friction factor becomes independent of Reynolds number. For comparison, in pipe flow, fully developed turbulent flow with a constant friction factor occurs at $Re \sim 10,000$ in very rough pipes (relative roughness $> 0.02$).

Internal melt generation is calculated through an energy balance at the bed:

$$\dot{m} = \frac{1}{L}(G + |\boldsymbol{u_b} \cdot \boldsymbol{\tau_b}| - \rho_w g \boldsymbol{q} \cdot \nabla h - c_t c_w \rho_w \boldsymbol{q} \cdot \nabla p_w) \tag{9}$$

where $L$ is latent heat of fusion of water, $G$ is geothermal flux, $\boldsymbol{u_b}$ is the ice basal velocity vector, $\boldsymbol{\tau_b}$ is the stress exerted by the bed onto the ice, $c_t$ is the change of pressure melting point with temperature, and $c_w$ is the heat capacity of water. Melt is therefore produced through a combination of geothermal flux, frictional heat due to sliding, and heat generated through internal dissipation (where mechanical kinetic energy is converted to thermal energy), minus the heat consumed or released in maintaining the water at the pressure-melting temperature in the presence of changing water pressure. We note that this form of the energy equation assumes that all heat produced is converted locally to melt and neglects advective transport and storage of dissipative heat. We assume that the ice and liquid water are isothermal, consistently at the pressure melting point temperature. These assumptions may not be strictly valid under certain real conditions that may have interesting heat transfer implications, such as heat advection (Clarke, 2003) or supercooling (Creyts and Clarke, 2010), or where meltwater enters a system of cold ice (below the pressure melting point), but we leave these potential model extensions for future work. As mentioned previously in Section 1.2, Werder et al. (2013) and similar models do not include the internal dissipation term in their sheetlike drainage components, but assign any melt from dissipation to contribute to opening in the nearest channel component.

For the sake of versatility, we also include an option to parameterize storage in the englacial system (note that this is not necessary for numerical stability; we use zero englacial storage in the example simulations presented in Section 3 of this paper). Following Werder et al. (2013), the englacial storage volume is defined as a function of water pressure:

$$b_e = e_v \frac{\rho_w g h - \rho_w g z_b}{\rho_w g} = e_v(h - z_b) \tag{10}$$

where $e_v$ is the englacial void ratio ($e_v = 0$ for no englacial storage).

Equations (1), (2), (5), and (9) are combined to form a parabolic, nonlinear partial differential equation (PDE) in terms of hydraulic head, $h$:

$$\nabla \cdot \left[\frac{-b^3 g}{12\nu(1 + \omega Re)} \nabla h\right] + \frac{\partial e_v(h - z_b)}{\partial t} = \dot{m}\left[\frac{1}{\rho_w} - \frac{1}{\rho_i}\right] + A|p_i - p_w|^{n-1}(p_i - p_w)b - \beta u_b + i_{e \to b} \tag{11}$$

With no englacial storage ($e_v = 0$), Eq. (11) takes the form of an elliptic PDE.

Defining a hydraulic transmissivity tensor:

$$\boldsymbol{K} = \frac{b^3 g}{12\nu(1+\omega Re)}\mathbf{I} \tag{12}$$

Equation (13) can be written more compactly as:

$$\nabla \cdot (-\boldsymbol{K} \cdot \nabla h) + \frac{\partial e_v(h - z_b)}{\partial t} = \dot{m}\left(\frac{1}{\rho_w} - \frac{1}{\rho_i}\right) + A|p_i - p_w|^{n-1}(p_i - p_w)b - \beta u_b + i_{e \to b} \tag{13}$$

Although we employ an isotropic representation of the hydraulic transmissivity tensor in Eq. (12), our model formulation can be readily generalized to incorporate anisotropy. The source terms on the right side of the Eq. (13) and the conductivity depend on $h$, as a result of which Eq. (13) is nonlinear, and solving for $h$ requires iterative methods.

## 2.2  Boundary conditions

Boundary conditions can be applied as either prescribed head (Dirichlet) conditions or as flux (Neumann) conditions. To
represent land-terminating glaciers, we typically apply a Dirichlet boundary condition of atmospheric pressure at the edge of the ice sheet:

$$h_{front} = z_b \tag{14}$$

To represent marine terminating glaciers, the outlet boundary condition can be set to the overlying fjord water pressure. Prescribed flux boundary conditions are imposed on the other boundaries of the subglacial drainage domain:

$$\nabla h_{bound} = f \tag{15}$$

where $f$ can be set to represent no flux ($f = 0$) or a prescribed flux, which can be constant or time-varying.

In our current formulation, there is no lower limit imposed on the water pressure; this means that unphysical negative pressures can be calculated in the presence of steep bed slopes, as in Werder et al. (2013). While suction and cavitation may occur in these situations, the flow most likely transitions to free-surface flow with the subglacial gap partially filled by air or
water vapor. At high water pressure, we restrict the value to not exceed the ice overburden pressure, which would in reality manifest as uplift of the ice or hydrofracturing at the bed. These extreme "underpressure" and "overpressure" regimes are important situations that have been considered in other studies (e.g., Tsai and Rice, 2010; Hewitt et al., 2012; Schoof et al., 2012), but are quite complex in 2D, and remain to be addressed carefully in future developments.

## 2.3  Computational strategy and implementation in the Ice Sheet System Model (ISSM)

The overall computational strategy employed is semi-implicit with an implicit backward Euler discretization of Eq. (13) to solve for the head field ($h$), combined with an explicit treatment of Eq. (2) for the evolution of the gap height ($b$). Within each

time step, the nonlinear Eq. (13) is solved using Picard iteration to obtain the head ($h$) field. From $h$, we calculate $p_w$, $\boldsymbol{q}$, $Re$, and $\dot{m}$, to be used in the subsequent iteration (in each iteration, $p_w$, $\boldsymbol{q}$, $Re$, and $\dot{m}$ are lagged from the previous iteration). Once the Picard iteration has successfully converged to a solution for $h$, the gap height geometry ($b$) is then updated explicitly based on basal gap dynamics using Eq. (2) to advance to the next time step. A schematic of this numerical procedure is presented in

Fig. 1. Due to the explicit treatment of Eq. (2), there is a time step limitation, which will be discussed further in Section 4.

SHAKTI is implemented within ISSM, an open source ice dynamics model for Greenland and Antarctica developed by NASA's Jet Propulsion Laboratory and University of California at Irvine (Larour et al., 2012; http://issm.jpl.nasa.gov). ISSM uses finite element methods and parallel computing technologies, and includes sophisticated data assimilation and sensitivity analysis tools, to support numerous capabilities for ice sheet modeling applications on a variety of scales. The SHAKTI

hydrology model solves the equations presented above in a parallel architecture using linear finite elements (i.e. P1 triangular Lagrange finite elements), which can be based on a structured or unstructured mesh. The source code is written in C++ and we rely on data structures and solvers provided by the Portable, Extensible Toolkit for Scientific Computation (PETSc, http://www.mcs.anl.gov/petsc). The user interface in MATLAB is the same as for other solutions implemented in ISSM, designed to facilitate model set up and post processing (see Documentation, https://issm.jpl.nasa.gov/documentation/hydrologyshakti/).

The iterative solution of Eq. (13) for hydraulic head employs the direct linear solver MUMPS in PETSc in each iteration, but other solvers provided by PETSc could be easily tested in future work.

Model inputs include spatial fields of bed elevation, ice surface elevation, initial hydraulic head, initial basal gap height, ice sliding velocity, basal friction coefficient, typical bed bump height and spacing, englacial input to the bed (which can be constant or time-varying, and can be spatially distributed or located at discrete points to represent moulin input), and appropriate

boundary conditions. Parameters that can either be specified or rely on a default value are geothermal flux, the ice flow law parameter and exponent, and the englacial storage coefficient.

Model outputs include spatiotemporal fields of hydraulic head, effective pressure, subglacial gap height (the effective geometry representative of an entire element), depth-integrated water flux, and "degree of channelization" (the ratio of opening by melt in each element to the total rate of opening in that element by both melt and sliding). Head and effective pressure are

calculated at each vertex on the mesh; gap height, water flux, and degree of channelization are calculated over each element (these quantities are based on the head gradient). Instructions for setting up, running a simulation, and plotting outputs can be found in the SHAKTI model documentation (https://issm.jpl.nasa.gov/documentation/hydrologyshakti/) and in an example tutorial (https://issm.jpl.nasa.gov/documentation/tutorials/shakti/).

## 3   Application

To demonstrate the capabilities of SHAKTI, here we present simple illustrative simulations that highlight some of its features. These test problems are designed to show the formation of sheetlike and channelized drainage in the context of different input scenarios (steady input, transient input, moulin point inputs, and distributed input) in simple model domains. We explore mesh

dependence of the model for the more complex examples in Sections 3.2 and 3.3, with further discussion of this and other limitations included below in Section 4.

## 3.1 Channel formation from discrete moulin input

In this first example, we consider a 1 km square, 500 m thick tilted ice slab with surface and bed slope of 0.02 along the $x$ direction. Steady input of 4 m$^3$ s$^{-1}$ is prescribed at a single moulin at the center of the square ($x = 500$ m, $y = 500$ m). Water pressure at the outflow (left edge of the domain, $x = 0$) is set to atmospheric pressure, with zero flux boundary conditions at the other three sides of the domain. All other constants and parameters are as described in Table 2. We use an unstructured triangular mesh with typical edge length of 20 m (with 4,004 elements). The model is run to a steady configuration (steady state is reached by 12 days) starting from an initial gap height of 0.01 m. A channelized drainage pathway emerges from the moulin to the outflow, with higher effective pressure (i.e. lower head and water pressure), larger gap height, and higher basal flux than its surroundings (Fig. 2). The degree of channelization metric also indicates a value close to 1 (indicating that opening by melt dominates opening by sliding) within the channelized drainage path. Note that the precise configuration of the channelized pathway is somewhat influenced by the unstructured mesh. Mesh sensitivity will be examined below in Section 3.2.

Scripts for running this example are included as a tutorial in ISSM (https://issm.jpl.nasa.gov/documentation/tutorials/shakti/), and can serve as a template for more sophisticated simulations. Run times will vary by machine and number of processors, but to run this simulation on 24 processors for 30 days with a time step of 1 hour, the entire simulation has a run time of approximately 38 seconds.

## 3.2 Channelization with multiple moulins

For the next example, we consider a rectangular domain 10 km long and 2 km wide, with a flat bed ($z_b = 0$ everywhere) and parabolic surface profile with a minimum thickness of 300 m and a maximum of 610 m. Ten moulins are located at arbitrarily chosen locations in the domain, each with a steady input of 10 m$^3$ s$^{-1}$. The model is run to 365 days with a time step of 1 hour (steady state is reached before 50 days), starting from an initial gap height of 0.01 m. The resulting steady distributions shown in Fig. 3 on five different meshes show a clear channelized drainage structure. Rather than each moulin forming a unique channel to the outflow, the moulin inputs influence each other, warping the pressure field and forming arborescent efficient pathways that combine downstream. For this specific arrangement of moulin inputs, a single principal drainage channel emerges. The unique drainage configuration that evolves in a particular circumstance and setting is affected by many factors, including bed topography, ice thickness, sliding velocity, meltwater input location, and input intensity.

The exact configuration of self-organizing channels also depends to some extent on the mesh. The five unstructured meshes used in this example have typical edge lengths ranging from 50 m (12,714 elements) to 400 m (205 elements). Using an unstructured mesh reduces bias in channel direction compared to a structured mesh, but the orientation and size of the elements does still affect the resulting geometry. Most subglacial hydrology models that resolve individual channels are mesh-dependent (e.g., Werder et al., 2013). The different cases shown in Fig. 3 provide a qualitative view of dependence of channelization structure on mesh size. Specifically, the gap height field on the coarsest mesh does not show a clear channel, and a well-

defined narrow channel is evident for larger distances upstream from the outflow boundary as the mesh is refined. The general structure of the channel is quite similar in the two finest meshes, but differences in alignment persist due to the unstructured nature of the mesh. From the viewpoint of coupling to ice motion and sliding calculations, the subglacial head and effective pressure fields obtained from the subglacial hydrology model are most important. The head and effective pressure fields shown in Fig. 3 are much smoother than the gap height field, and appear to show less sensitivity to the mesh size. To evaluate this sensitivity further, Fig. 4 presents quantitative plots of the mean head and effective pressure (averaged in the $y$ direction) for the five meshes. Across much of the domain, they converge remarkably well, but diverge slightly in the region of significant channelization.

## 3.3   Seasonal variation and distributed meltwater input

Next we consider a transient example involving a seasonal input cycle of meltwater, with input distributed uniformly across a rectangular domain 4 km long and 8 km wide. The bed is flat ($z_b = 0$ everywhere). The ice surface follows a parabolic profile, with ice thickness ranging from 550 m at $x = 0$ to 700 m at $x = 4$ km, and is uniform across the $y$ direction. We begin with an initial subglacial gap height of 0.01 m, perturbed with random variations drawn from a normal distribution with standard deviation of 1%. The purpose of these random variations in the initial gap height is to serve as triggers for potential instability and channelization, which is an important phenomenon in subglacial hydrologic systems (Walder, 1986; Kamb, 1987; Schoof, 2010; Hewitt et al., 2011). Even in nature, the gap height is unlikely to be uniform and the ubiquitous irregular variations in the gap height and bedrock surface will act as natural perturbations to initiate instabilities and channelization. As the ice slides over bedrock, abrasion processes may also serve to generate irregularities. In the literature on the self-organized formation of dissolution channels in rock fractures in karst formations (e.g. Cheung and Rajaram, 2002; Szymczak and Ladd, 2006; Rajaram et al., 2009), it has been established that under conditions that lead to self-organized channel formation, the specific nature of the initial random variations do not influence the structure and spacing of the channels; rather they serve as a trigger for the initiation of channels. In unstructured meshes, it is also possible for mesh-related asymmetries to introduce perturbations that can serve as triggers for this instability. In stable regimes, however, the same perturbations will not produce channelization.

The model is first run with steady distributed input of 1 m a$^{-1}$ in a spin-up stage with a time step of 1 hour (steady state achieved in 4 days). After a steady configuration is achieved, a cycle of meltwater input variation is imposed and run for 1 year (365 days), also with a time step of 1 hour. Seasonal meltwater input in m a$^{-1}$ is approximated by a cosine function between 0.4-0.7 a (days 146 and 255):

$$i_{e \to b} = -492.75 \times \cos(2\pi/0.3(t - 0.4)) + 493.75 \tag{16}$$

This yields a maximum meltwater input at the peak of the summer of 986 m a$^{-1}$, with a winter minimum of 1 m a$^{-1}$, and annual mean input of 149 m a$^{-1}$. The peak melt input corresponds to approximately 1,000 m$^3$ s$^{-1}$ for the entire domain. Note that the values used here are unrealistically high, and are designed intentionally to show stable behavior of the system across a variety of input magnitudes, even when subjected to extreme forcing. Figure 5 shows time series plots of this "seasonal" input

forcing over one full annual cycle, with the corresponding minimum, mean, and maximum gap height and head. Snapshots of the subglacial hydrologic variable fields at intervals through the annual cycle are shown in Fig. 6, and an animation of this simulation is included in the supplementary material. As melt increases, the maximum gap height increases, corresponding to growth of the subglacial system and emergence of self-organized efficient channels. The maximum gap height increases with

increasing meltwater input until the peak of the melt season, then decreases simultaneously as melt input decreases (note that we use zero englacial storage in this simulation, so there is no lag due to water storage in the system). The hydraulic head initially increases with increased input (meaning an increase in subglacial water pressure as additional water is added to the system), then decreases as efficient low-pressure channels form, then increases again as melt starts to decrease and the channels collapse. We hold the sliding velocity constant, but in reality ice sheet sliding velocity generally increases with increased water

pressure (i.e. lower effective pressure) and decreases with lower water pressure. With two-way coupling between the subglacial system and ice dynamics (e.g., Hoffman and Price, 2014; Koziol and Arnold, 2018), the sequence of hydraulic head or basal water pressure variation seen here would likely result in a mid-to-late summer decline in sliding velocity, after which the sliding velocity would increase again. Subsequently, as melt input decreases to the winter minimum, the hydraulic head decreases to low values, which would correspond to a decrease in sliding velocity. As shown in Fig. 6, for the early and late parts of the year,

the system essentially behaves as a one-dimensional system, because the melt inputs are not large enough to take the system into a regime where channelization can occur. During the melt season, when inputs increase substantially, self-organized, regularly spaced channels emerge, seen in Fig. 6 as having lower heads than their immediate surroundings in the $y$ direction. These channelized structures collapse and disappear entirely as the meltwater input drops off and returns to the winter minimum. The simulation results shown here demonstrate the ability of our modeling framework to represent both stable regimes, where the

subglacial system takes on a relatively smooth quasi-one-dimensional configuration, and unstable regimes with self-organized efficient pathways when high meltwater inputs and discharge trigger the transition to channelization.

     To examine mesh dependence in this case of self-organized channelization, Fig. 7 presents gap height and head distributions on three unstructured meshes with typical edge lengths of 50 m, 100 m, and 200 m. At 100 m resolution, the channelization effects are obvious, with similar spacing as on the finer 50 m mesh. At 200 m resolution, the channels are still apparent but

the head and effective pressure fields are more smoothed than with the finer meshes, especially in the upstream portions of the domain. In the early and late parts of the cycle, the behavior obtained with different mesh sizes are in good agreement for sheetlike drainage. The mesh dependence is evaluated more quantitatively in Fig. 8 with $y$-averaged quantities for for Day 1 (sheetlike drainage everywhere), Day 200 (peak melt input and extreme channelization), and Day 250 (near the end of the melt input cycle as channelization collapses). We see that the solutions obtained with different mesh resolutions converge well

for sheetlike drainage, but they show some variation with channelization. These local differences are more pronounced in the quantities calculated over elements (gap height and degree of channelization), while differences are relatively small in the smooth pressure distributions calculated at mesh vertices.

## 4 Discussion

The flexible geometry and flow regimes of the SHAKTI model allow for various drainage configurations to arise naturally. We conserve mass and energy in all parts of the domain, in contrast to several existing models that neglect the role of melt opening in sheetlike drainage systems or redistribute dissipated mechanical energy in the sheet system to adjacent channels. Previous studies found that with similar equations, including the melt term in a distributed system leads to an instability and runaway growth, which initiates channelization (Schoof, 2010; Hewitt, 2011). In our formulation, even including melt from internal dissipation, we are able to achieve stable configurations of subglacial geometry, basal water flux, and pressure fields with steady and transient input forcing. Channelized pathways with lower water pressure than their surroundings form from moulin inputs (Figs. 2 and 3) as well as self-organized configurations with high distributed melt input (Fig. 6). A feature of our formulation that contributes to this behavior is the way we calculate the basal water flux (approximate momentum equation, Eq. 5), which allows for a transient, spatially variable transmissivity that transitions naturally between laminar and turbulent flow regimes locally, while allowing both types of flow regime to coexist in the model domain, as well as flow that exhibits attributes along the wide transition between laminar and turbulent flow. To illustrate this behavior more clearly, Fig. 9 presents the distribution of Reynolds number through the initiation of channelization for days 145-175 of the transient example in Section 3.3. On Day 145 (just before the onset of increased melt input, see Fig. 5), the Reynolds number is low throughout the domain (the maximum Reynolds number is only about 70), corresponding to laminar flow. On Day 155, Reynolds number has increased, particularly near the outflow at the left, transitioning into the turbulent regime in much of the domain with $Re > 1,000$. As the self-organized channelized structure emerges through Days 165 and 175, Reynolds number becomes increasingly higher in the channelized pathways than their surroundings. If we were to use a purely laminar or purely turbulent flux formulation, the nature of the flow and the mechanical energy dissipation rate would not be accurately represented across this range of Reynolds numbers. If the flux is simulated as laminar everywhere (using a very small value of $\omega$ in Eq. (5), so that $\omega Re << 1$ and the flux is always linearly proportional to the head gradient), channelization does still occur with high inputs, but the flow mechanics are not correctly represented for regions with large Reynolds number. If we force the flux to be turbulent everywhere (by using a large value for $\omega$ in Eq. 5, so that $\omega Re >> 1$ and the flux is always proportional to the square root of the head gradient), the nonlinear iteration to solve Eq. (15) encounters non-convergence with large oscillations between Picard iterations for the same model problems that behave well when we employ the flux Eq. (5), which allows for laminar, transitional, and turbulent flow regimes. The concept of laminar-turbulent transition is well established in hydraulics and fluid mechanics, and our representation of the nonlinear flux-gradient relationship (Eq. 5) is consistent with this concept and is also consistent with experimental studies of Zimmerman et al. (2004) on rock fractures with non-smooth walls.

The transient example in Section 3.3 illustrates one possible pattern of idealized seasonal evolution of the subglacial drainage system, where channels emerge with increased melt and collapse to a sheetlike system again in the winter. The higher water pressure during the melt season would imply increased sliding velocity in a two-way coupled system, with a decrease in mid-to-late summer with well established channelized drainage, followed by an increase as the efficient system initiates its shutdown, and a decrease as meltwater input returns to the background winter rate. This seasonal pattern is reminiscent of

observations of some Greenland outlet glaciers (Moon et al., 2014), and subglacial hydrology may indeed play a key role in shaping the seasonal velocity behavior of some glaciers, both land-terminating and marine-terminating. In future work on real glacier topography, we aim to investigate other velocity signatures, such as those that experience an annual minimum velocity in the late melt season, which is thought to be a result of highly efficient channel development (Moon et al., 2014) or those

5 with high winter sliding velocities, which may be indicative of hydraulically isolated or poorly connected regions of the bed that maintain high water pressure through winter (e.g., Hoffman et al., 2016; Downs et al., 2018; Rada and Schoof, 2018). To accurately capture the influence of transient sliding velocities on the evolution of subglacial hydrology, two-way coupling between subglacial hydrology and ice dynamics is important.

## 4.1 Model limitations

This paper is intended to present a description of the SHAKTI model formulation with illustrative simulations under simple scenarios. Application to real glaciers remains for upcoming work, but we wish to clearly address limitations of the model and acknowledge challenges faced by this and other subglacial hydrology models.

Time-stepping is an important factor in numerical models of the highly transient subglacial hydrologic system, such as SHAKTI. To illustrate the influence of time step size, Fig. 10 presents evolution of maximum head in the single moulin ex-

15 ample (see Section 3.1 and Fig. 2) for different time step sizes. In this example, the model converges properly to the same steady configuration for time step sizes dt=0.25 h to dt=3 h. Note that as the time step increases to about 3 h, small but stable fluctuations are seen. With dt=4 h, however, the model never converges to the solution, but instead enters a large systematic oscillation between incorrect values. For larger time steps than dt=4 h, the nonlinear iteration itself has difficulty converging and the amplitude of the oscillations becomes very large with water pressure exceeding ice overburden pressure, which is

20 accompanied by very large dissipation rates. Difficulties in convergence during numerical solutions of nonlinear PDEs with larger time-steps is a well-known issue in a variety of contexts. The appropriate time step size is dependent on various parameters specific to a simulation such as topography, ice thickness, and meltwater input rates. Due to the highly nonlinear nature of the equations, it is unfortunately not straightforward to establish a time step criterion for stable model behavior. As a general guideline we suggest conducting an initial test with a time step of 1 hour and adjusting accordingly. We plan to implement

adaptive time-stepping in future developments of SHAKTI. Note that the time steps required in subglacial hydrology models are typically much smaller than time steps frequently used in long-term ice dynamics simulations, which may be on the order of years or decades. Although it is desirable to maintain longer time steps in subglacial hydrology models, the essential physics operates on much smaller time scales and using a smaller time step of the order of hours may be unavoidable. Coupling with ice sheet models may rely on spatio-temporally integrated basal water and effective pressures.

We calculate basal gap height over each element, which means that the geometry is dependent on mesh size. It is not our aim to necessarily capture each individual cavity or channel cross-section, but rather to obtain the effective geometry over each element and its effect on the pressure field, which has an important influence on ice sheet sliding velocity. In Sections 3.2-3.3, we examined mesh sensitivity in example simulations (see Figs. 3 and 7). With very large elements (km scale), the effects of channelized drainage may be smoothed out. For large-scale simulations, a variable mesh should be used with coarser resolution

in the ice sheet interior away from the margins and finer resolution at lower elevations where the bulk of meltwater is produced and enters the subglacial system (where channelized networks are likely to form and sliding velocities are higher). The typical edge length scale should be selected according to the particular application, depending on the resolution of bed topography, sliding velocities, modeling goals, as well as practical concerns of computing power. As a rough guideline, to capture the formation of channelization in decent detail, we suggest an edge length of 150 m or less in the domain area of most interest (e.g., the few km nearest the terminus of a glacier).

As stated in Section 2.2, the current formulation does not handle either high water pressures that exceed overburden (we cap water pressure at overburden pressure and do not represent uplift) or low water pressures where the system would transition to free surface flow (we assume the subglacial gap is always filled with water and allow unphysical negative water pressures to be calculated in the presence of steep slopes). The sample simulations presented in Section 3 do not involve either of these extreme pressure ranges in their solutions, so the results included here are unaffected by the upper limit imposed on water pressure or by allowing negative water pressures in lieu of transitioning to a partially filled system.

The examples in Section 3 do not involve complex bed topography, which is beyond the scope of this initial model description paper. The model has been successfully tested on real ice and bed geometry, however, and results will be included in forthcoming work.

Under thick ice with low meltwater input, the nonlinear iteration may have trouble converging to a head solution, entering a stable oscillation. This can frequently be resolved by decreasing the time step and/or employing under-relaxation to help the nonlinear iteration converge.

The SHAKTI model is not currently coupled to ice dynamics in a two-way manner. We prescribe a constant ice sliding velocity, and this sliding velocity does not evolve according to the influence of subglacial water pressure. With this one-way coupling, we are able to infer only qualitatively how the ice velocity would be affected by the changing subglacial system. In upcoming work, we plan to implement two-way coupling with the ice dynamics of ISSM to test different sliding laws and the behavior of the fully coupled system.

## 5  Conclusions

In this paper, we presented the SHAKTI model formulation with simple illustrative simulations to highlight some of the model features under different conditions. The model is similar to previous subglacial hydrology models, but employs a single set of "unified" governing equations over the entire domain, including opening by melt from internal dissipation everywhere, without imposing a distinction between channelized or sheetlike systems. The geometry is free to evolve; efficient, low-pressure channelized pathways can and do form as the subglacial system adjusts and facilitates transitions between different flow regimes. We find that with high meltwater input (via moulins or distributed input), self-organized channelized structures emerge with higher effective pressure (i.e. lower water pressure) than their surrounding areas. As meltwater input decreases, these channelized drainage structures collapse and disappear.

To understand the overall mass balance and behavior of glaciers and ice sheets, it is crucial to understand different observed seasonal velocity patterns, and the corresponding enigmatic drainage systems hidden beneath the ice. Combined with advances in remote and field-based observations, and modeling of other processes involved in the hydrologic cycle of ice sheets and glaciers (such as surface mass balance, meltwater percolation and retention, and englacial transport of water), subglacial hydrology modeling may help close a gap in ice dynamics models to inform predictions of future mass loss and sea level rise. Forthcoming work will focus on application of the SHAKTI model to real glaciers and coupling the model to an ice dynamics model (ISSM, into which SHAKTI is already built).

*Code availability.* The SHAKTI model is freely available as part of the open source Ice Sheet System Model (ISSM), which is hosted in a subversion repository. https://issm.jpl.nasa.gov/download/

*Author contributions.* HR and AS formulated the model equations. AS wrote the stand-alone versions of the finite volume and finite element models. MM built the parallel model into ISSM and assisted AS with further model development. AS performed simulations and compiled the manuscript with contributions from HR and MM.

*Competing interests.* The authors declare that they have no conflicts of interest.

*Acknowledgements.* This work was primarily supported by a NASA Earth and Space Science Fellowship award (NNX14AL24H) to AS. A version of this model was originally presented in a 2010 proposal by HR and Robert Anderson. We thank Robert Anderson for his continued encouragement. Special thanks to Matthew Hoffman for many helpful conversations about subglacial hydrology modeling, to Basile DeFleurian and Mauro Werder for including our model in the Subglacial Hydrology Model Intercomparison Project (SHMIP, https://shmip.bitbucket.io/) and providing useful insights along the way, and to Eric Larour for his initial enthusiasm that facilitated our collaboration with ISSM. We also thank two anonymous reviewers for their constructive comments to improve the clarity and strength of this manuscript.

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

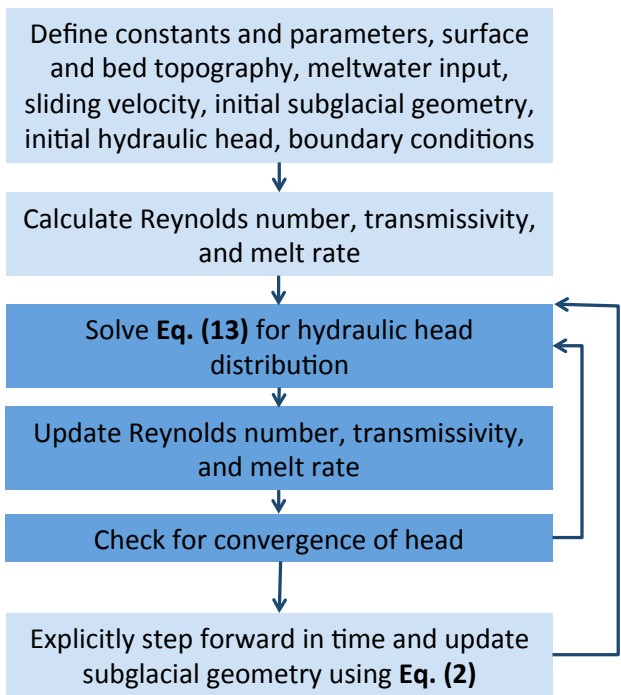

**Figure 1.** Schematic of computational procedure used to solve the model equations

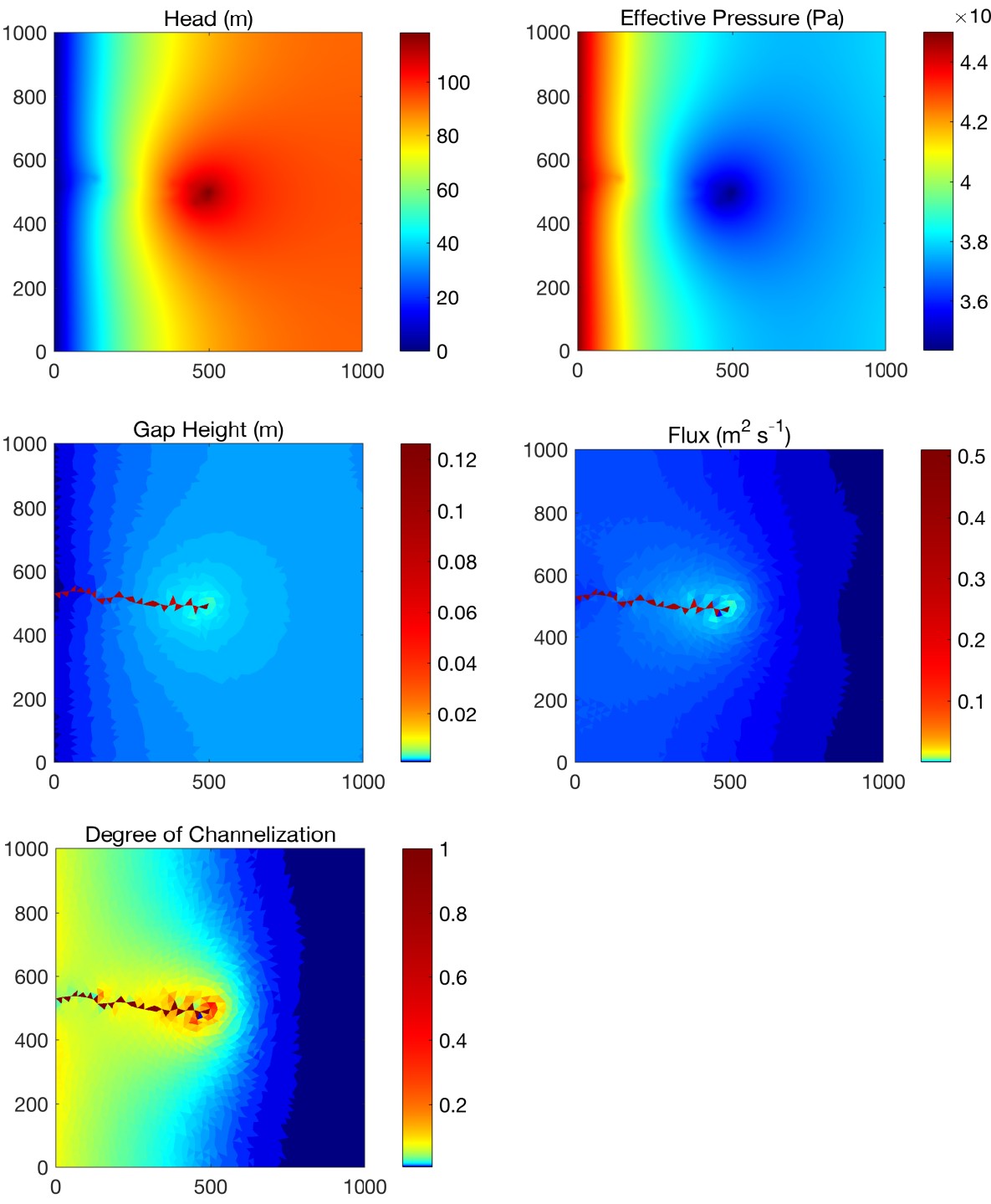

**Figure 2.** Steady configurations of hydraulic head, effective pressure, gap height, depth-integrated basal water flux, and degree of channelization for steady input of 4 m$^3$ s$^{-1}$ into a moulin at the center of a 1 km square domain. Ice thickness is 500 m, with surface and bed slope of 0.02. A clear efficient pathway forms from the moulin input to the outflow at the left edge of the domain.

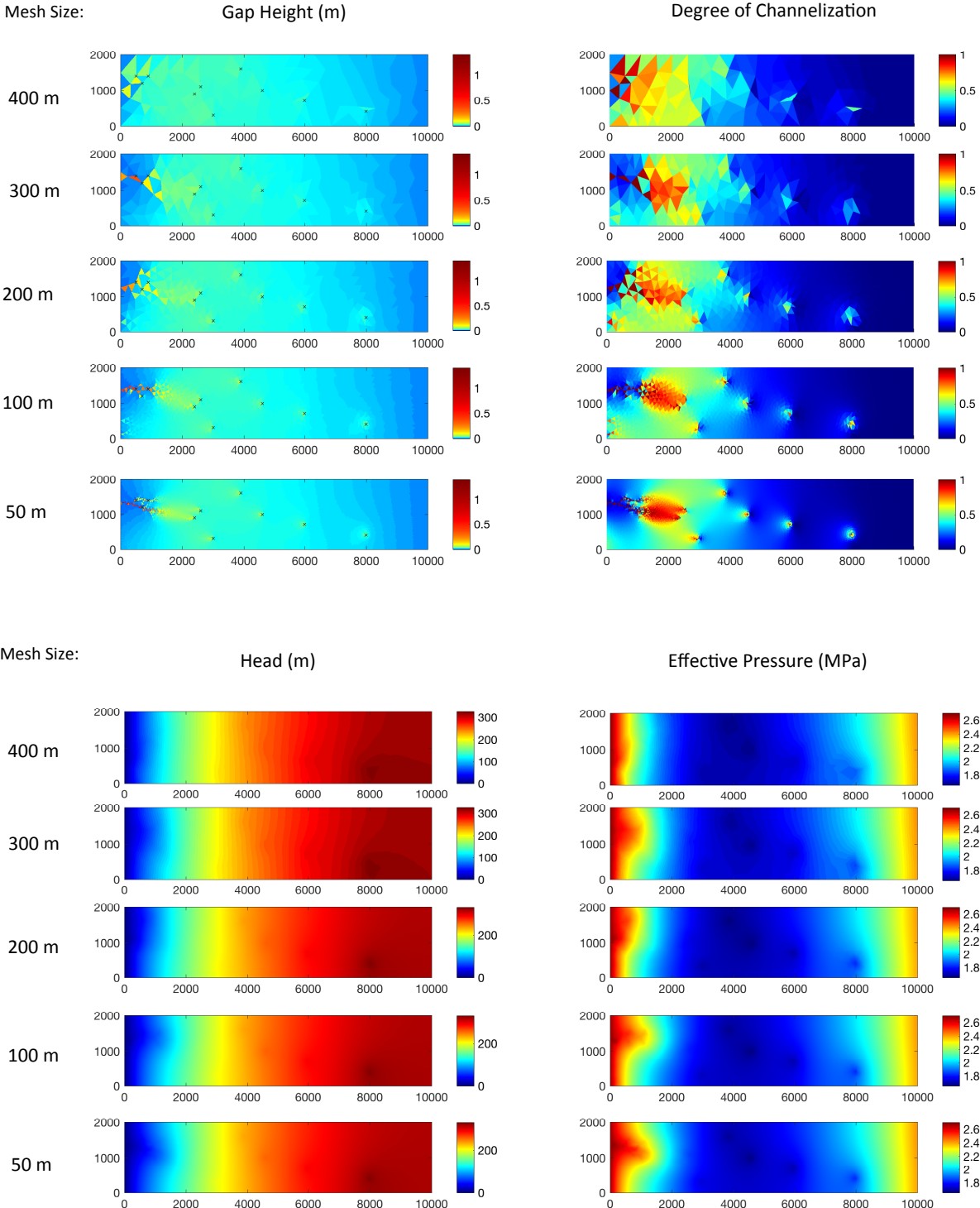

**Figure 3.** Steady-state distributions resulting from steady input of $10 \, \text{m}^3 \, \text{s}^{-1}$ into 10 moulins. As a qualitative evaluation of mesh dependence, results are shown for typical element side lengths ranging from 50 m to 400 m. Moulin locations are indicated on the gap height plots as black markers. Rather than each moulin forming an independent channel, the various inputs warp the pressure field and interact to produce a principal efficient drainage pathway.

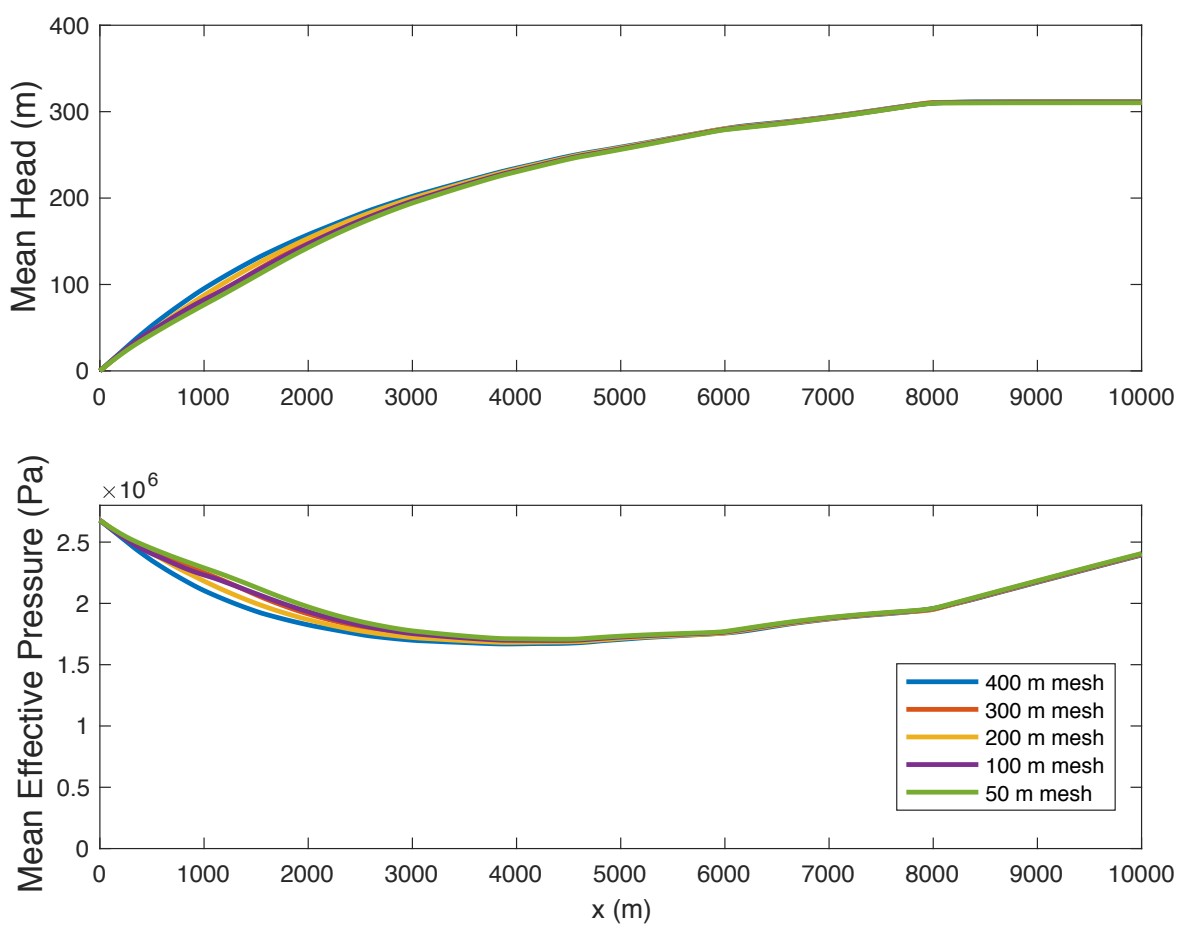

**Figure 4.** Mean head and effective pressure (averaged in y direction) for the 10-moulin example (Fig. 3) using unstructured meshes with typical element side lengths ranging from 50 m to 400 m.

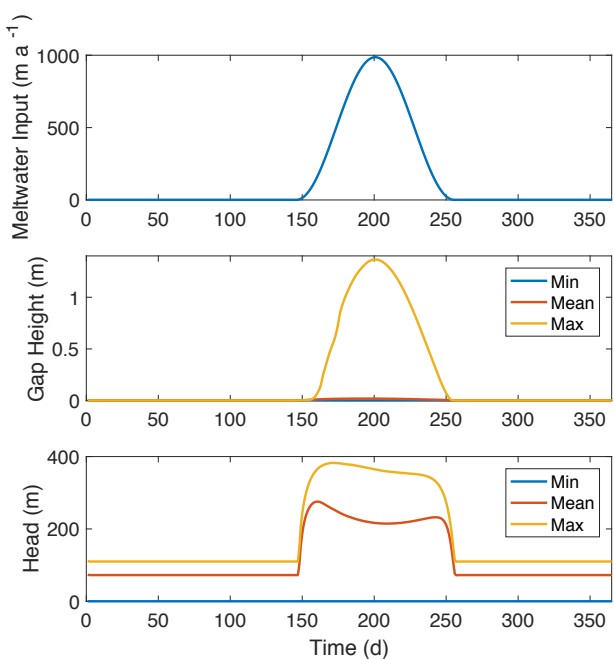

**Figure 5.** Seasonal cycle of distributed meltwater input over one annual cycle, with gap height and head evolution time series. As meltwater input increases, the maximum gap height increases, then decreases simultaneously with the decrease in input. As meltwater input increases, the head increases, then decreases as efficient drainage pathways are established (corresponding to lower water pressure in the efficient pathways, as well as lower head in the unchannelized upstream regions as shown in Fig. 6). As melt decreases, mean head increases again as the efficient pathways start to collapse, then decreases as melt returns to the winter minimum.

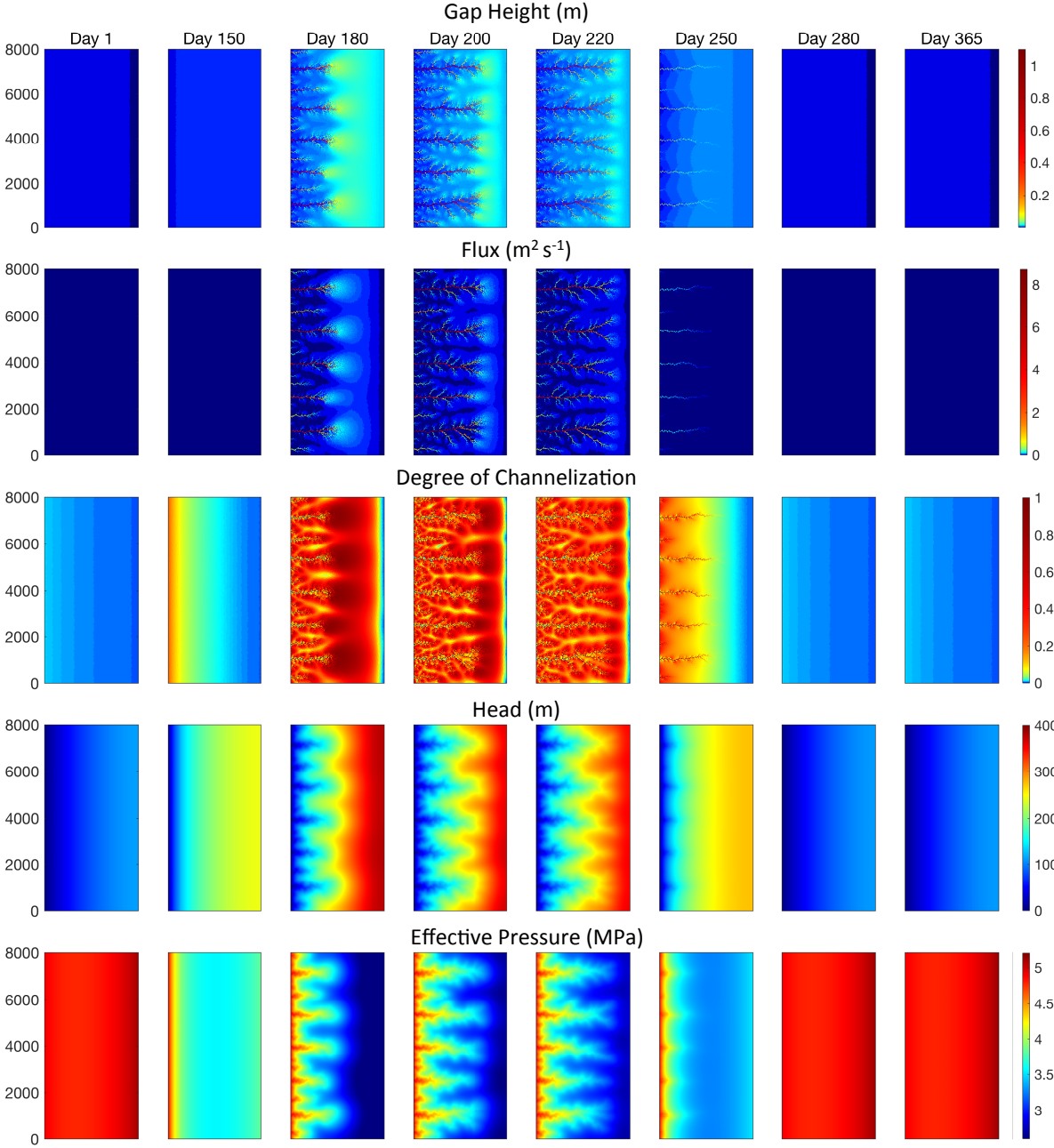

**Figure 6.** Seasonal evolution with distributed meltwater input as shown in Fig. 5 on a 4 km by 8 km domain over one full annual cycle. Self-organized efficient drainage pathways form from the outflow (left edge of the domain) as melt input increases, persist through the melt season, and collapse again as melt input decreases, returning to a steady sheet configuration. The efficient pathways show lower head (i.e. higher effective pressure) than their surrounding areas in the y direction.

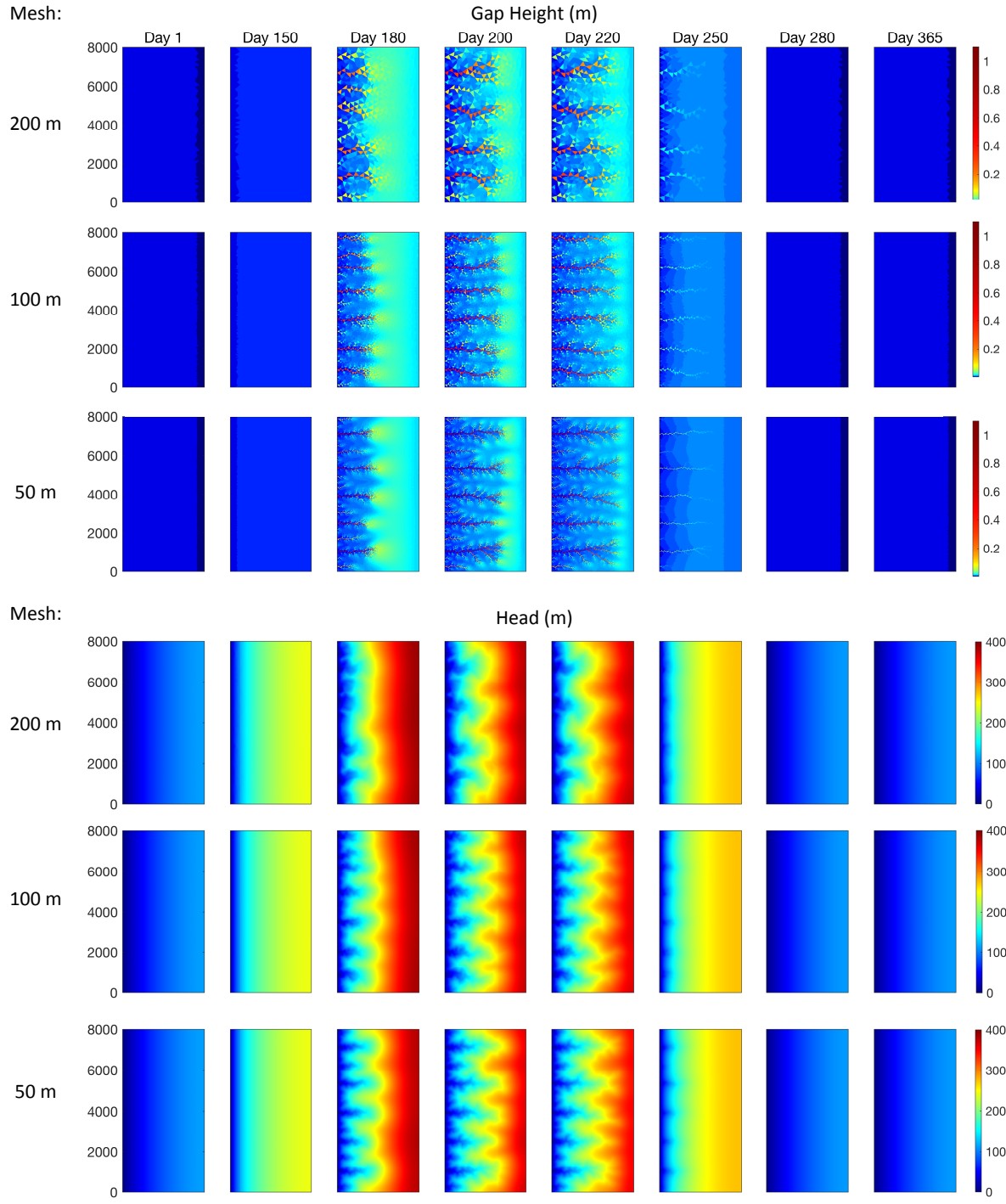

**Figure 7.** Mesh dependence shown for the transient example with distributed input (see Section 3.3 and Figs. 5 and 6) with typical element edge lengths of 50 m, 100 m, and 200 m.

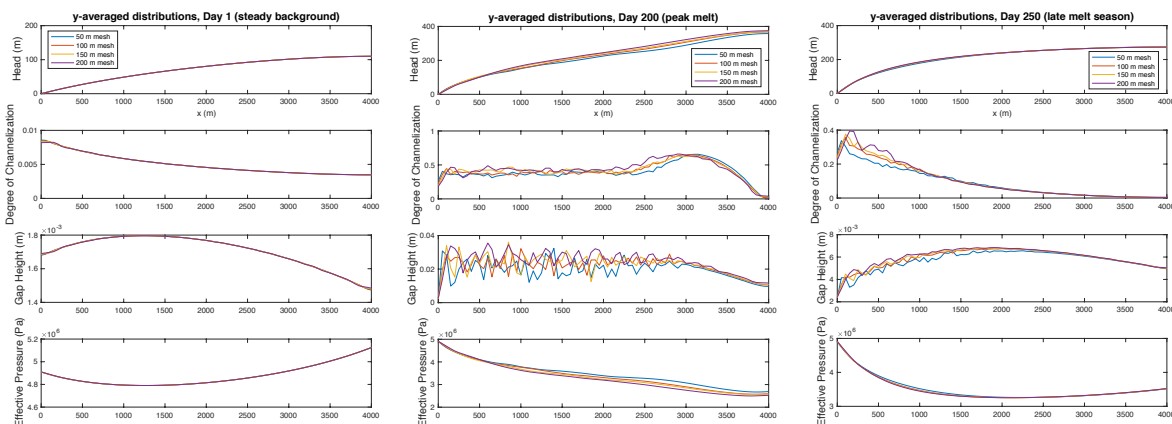

**Figure 8.** Mesh dependence shown with y-averaged quantities for the transient example (see Section 3.3 and Figs. 5-7) for three selected days. The model has very little dependence on mesh size with sheetlike drainage (Day 1). With channelization (Day 200 at the peak of the input and Day 250 with some channelization), mesh size leads to variability in the highly channelized regions. The local differences are more pronounced in the quantities calculated over elements (gap height and degree of channelization), while differences are relatively small in the smooth pressure distributions calculated at vertices of the mesh.

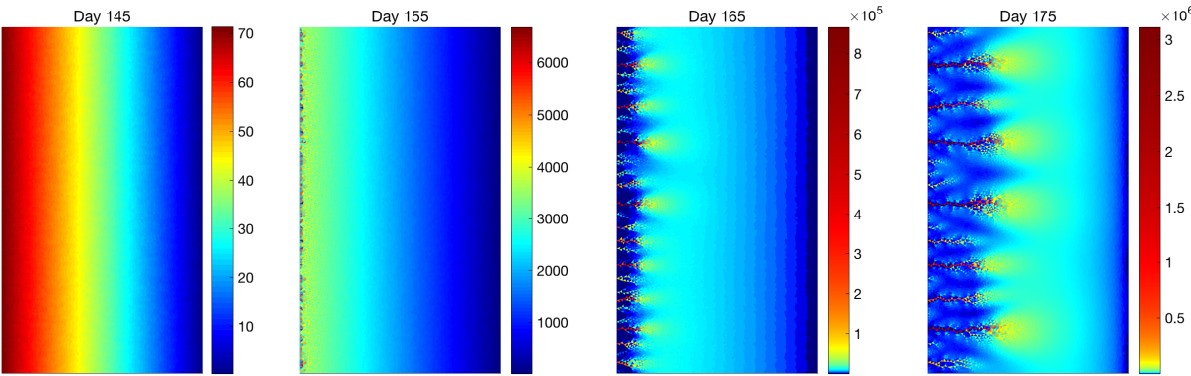

**Figure 9.** Reynolds number evolution during the onset of channelization in the transient example with distributed input (see Section 3.3 and Figs. 5 and 6). Initially, the entire domain has low Reynolds number, corresponding to laminar flow. As the meltwater input increases, Reynolds number transitions into the turbulent regime and becomes clearly higher in the self-organized channelized structures than in the surrounding sheetlike regions. Note that the color scale is different for each plot.

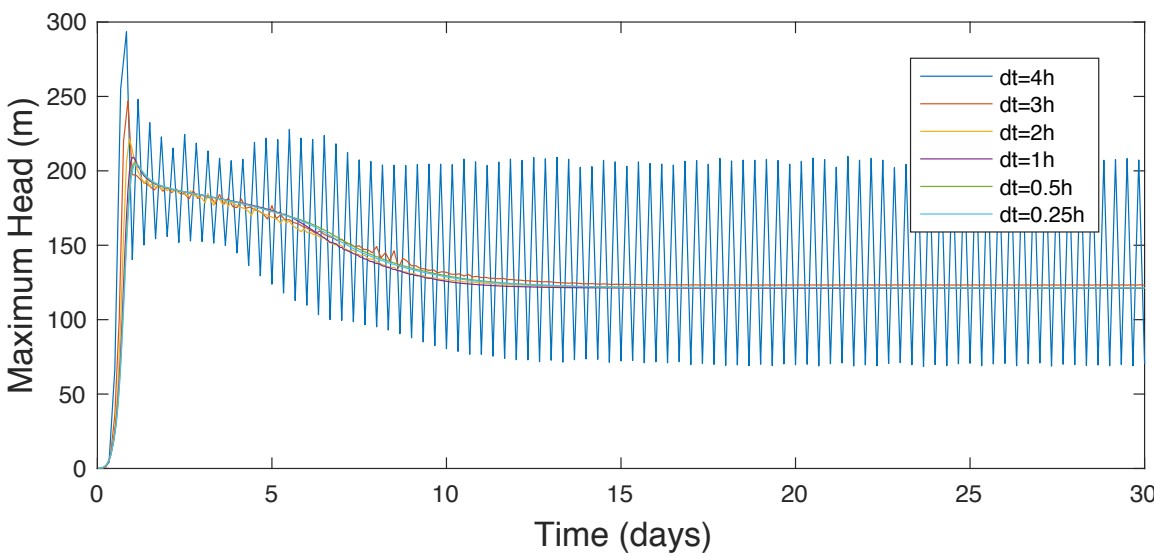

**Figure 10.** Maximum head evolution to illustrate time step dependence for the steady simulation with a single moulin input (see Section 3.1 and Fig. 2). For dt<4h, the model converges properly to the correct solution, but with dt=4h it enters a large, stable oscillation and never converges.

**Table 1.** Variables used in model equations

| Symbol | Units | Description |
|---|---|---|
| $b$ | m | Subglacial gap height (average over element) |
| $b_e$ | m | Englacial storage volume per unit area of bed, $b_e = e_v(h - z_b)$ |
| $t$ | s | Time |
| $\boldsymbol{q}$ | $\text{m}^2\,\text{s}^{-1}$ | Gap-integrated basal water flux, $\boldsymbol{q} = \frac{-b^3 g}{12\nu(1+\omega Re)}\nabla h$ |
| $\dot{m}$ | $\text{kg}\,\text{m}^{-2}\,\text{s}^{-1}$ | Internal melt rate |
| $p_i$ | Pa | Ice overburden pressure, $p_i = \rho_i g H$ |
| $p_w$ | Pa | Subglacial water pressure, $p_w = \rho_w g(h - z_b)$ |
| Re | Dimensionless | Reynolds number, $\text{Re} = |\boldsymbol{q}|/\nu$ |
| $h$ | m | Hydraulic head |
| $\beta$ | Dimensionless | Parameter to control opening due to sliding over bedrock bumps, $\beta = (b_r - b)/l_r$ for $b < b_r$, $\beta = 0$ for $b \geq b_r$ |
| $N$ | Pa | Effective pressure, $N = p_i - p_w$ |

**Table 2.** Constants and parameters

| Symbol | Value | Units | Description |
|---|---|---|---|
| $\rho_w$ | 1,000 | $\text{kg m}^{-3}$ | Bulk density of water |
| $i_{e \to b}$ | | $\text{m s}^{-1}$ | Input rate of meltwater from englacial system to subglacial system |
| $\rho_i$ | 910 | $\text{kg m}^{-3}$ | Bulk density of ice |
| $A$ | | $\text{Pa}^{-3} \text{ s}^{-1}$ | Flow law parameter |
| $n$ | 3 | Dimensionless | Flow law exponent |
| $b_r$ | 0.1 | m | Typical height of bed bumps |
| $l_r$ | 2.0 | m | Typical spacing between bed bumps |
| $u_b$ | $10^{-6}$ | $\text{m s}^{-1}$ | Sliding velocity ($31.5 \text{ m a}^{-1}$) |
| $g$ | 9.8 | $\text{m s}^{-2}$ | Gravitational acceleration |
| $\omega$ | 0.001 | Dimensionless | Parameter controlling nonlinear transition between laminar and turbulent flow |
| $L$ | $3.34 \times 10^5$ | $\text{J kg}^{-1}$ | Latent heat of fusion of water |
| $G$ | 0.05 | $\text{W m}^{-2}$ | Geothermal flux |
| $c_t$ | $7.5 \times 10^{-8}$ | $\text{K Pa}^{-1}$ | Change of pressure melting point with temperature |
| $c_w$ | $4.22 \times 10^3$ | $\text{J kg}^{-1} \text{ K}^{-1}$ | Heat capacity of water |
| $\nu$ | $1.787 \times 10^-6$ | $\text{m}^2 \text{ s}^{-1}$ | Kinematic viscosity of water |
| $e_v$ | | Dimensionless | Englacial void ratio |