# Peer review of "SHAKTI: Subglacial Hydrology And Kinetic, Transient Interactions v1.0"

_Geoscientific Model Development, 2018_

## Referee Comment (RC1) · Anonymous Referee #1 · 24 Apr 2018

The manuscript by Sommers et al. describes a subglacial hydrologic model which permits the representation of a range of turbulence regimes. The implementation of a spatially and temporally varying hydraulic transmissivity removes the need for individual representation of 'channelized' and 'distributed' model elements and allows for the inclusion of viscous dissipation across the model domain. This advance, modified from studies on water flow in rock fractures, is new to subglacial hydrologic models and could potentially represent the continuum of subglacial characteristics within a single model.

Overall, the relatively brief manuscript clearly lays out the model components and provides several idealized model experiments to characterize the system. The modeling results demonstrate that the implementation of this variable transmissivity results in

spatially and temporally varying subglacial pressures that qualitatively reproduce the expected seasonal evolution of the subglacial system. However, because the channelized component is represented on the model grid (not on grid edges as in previous modeling efforts), there is an unfortunate dependence of model output on the grid scale.

While this manuscript has strong points and is a novel extension of previous work, the key points of the model development effort are not strongly represented in the text. In particular, the text of the manuscript should better reflect both previous development of subglacial models and an improved description of the theoretical underpinnings. Details about this issue are indicated in the general comments and line comments below.

General comments

1. The introduction should be more clearly focused (and quite possibly expanded) on the topic of subglacial hydrology. There is a fairly extensive body of literature about subglacial model development, including extensive work on alpine glaciers. The focus on outlet glaciers and sea level rise in the introduction is somewhat of an aside.

2. The motivation of the manuscript is somewhat unclear if the reader is un-indoctrinated into the world of subglacial hydrology. It would be useful to include a through description of viscous dissipation and why it hasn't been included in previous subglacial models in section 1.2 and clearly describe - before the model description - the goals of this modeling effort.

3. The basal flux parameterization (Line 25) needs to be more carefully documented. There are several line notes to this effect, but essentially, the addition of the Reynolds number requires the selection of characteristic length scales and dimensionless parameters - reasoning behind how these values are assigned should be included in order to enhance the usefulness of this manuscript.

4. There should be a 'model limitations' section that includes a more through discussion of grid dimension sensitivity and perhaps a narrow sensitivity study on other parameters such as gap height initialization. In this section it might also be beneficial to include some discussion about how to determine the most appropriate grid scale - i.e. is there an ideal scale depending on ice thickness, roughness, coupling with an ice dynamics model, etc.

5. The citations/references should be checked. There are a few citations that are not in the references section and vice versa.

Specific comments Page 1 Line 1. I am not sure "poorly understood" is the best phrase to use here. There is an extensive body of literature exploring the state and evolution of the subglacial hydrologic system and its representation in current models, while not perfect, are able to replicate many features of ice velocity fields. We know that the link between melt and ice motion is the subglacial system; however, there are parameters and parameterizations which are not well constrained.

1-2. The wording of this sentence is awkward.

9. Much of the manuscript switches between 'channel' and 'efficient' drainage. Consider using something like '. . .over a wide range of drainage efficiencies. . .' or inefficient and efficient drainage to eliminate the "channel".

15-22. While understanding ice sheet dynamics is important for the characterization of future sea level rise, it might be more correct to acknowledge that basal lubrication alone may not be a major uncertainty in sea level rise predictions (e.g., IPCC, sea level change chapter, pages 1168-1169; Shannon et al., 2013).

21-22. Consider citing the chapter instead of the whole 'Physical Basis' document.

Page 2 7. There are a number of other publications that could be cited along with Cowton et al. (2013), including Bartholomew et al. (2010), Chandler et al. (2013), and Andrews et al. (2014).

[Figure]

8-9. This sentence seems out of place and doesn't provide much information.

12-13. It seems that a description of the unknowns would logically follow this sentence instead of a description of a description of how the subglacial system works. It might be useful to remove references to unknowns - while this is certainly true - the main purpose of the manuscript is to rectify a persistent known problem - that models of subglacial hydrology tend to only represent 2 endmembers of the continuum of possible configurations.

23-26. Reference the section numbers

27. This section would benefit from explaining the motivation for subglacial hydrology model development as well. See Flowers (2015) for a great review of the topic.

Page 3 3-4. This relates to the previous comment - it would be nice to detangle why the community ended up focusing on these two endmembers.

13. See comments regarding line p2L12-13 and p1L1.

34 - p4L5. These sentences start to feel rushed. Also consider including more recent work by Rada and Schoof (2018) and Downs et al. (2018).

Page 4 7-8. This sentence is a direct repeat of a sentence in the abstract. Consider revising.

13-15. The instability that arises with the viscous dissipation has been discussed by a number of studies (Hewitt et al., 2012; Hoffman and Price, 2014; Kamb, 1987; Schoof, 2010; Schoof et al., 2012; Walder, 1986; Werder et al., 2013). In addition, Flowers (2015) has a nice summary of the reasoning behind and numerical approaches to switching between drainage elements. Because the primary contribution of this work is the inclusion of the viscous dissipation term and the representation of both turbulent and laminar flow, it is important to thoroughly discuss the reasoning and justification and numerics used in previous modeling work. This summary could readily follow lines 13-15.

[Figure]

18. If the isolated/weakly connected system is the primary scientific motivation behind the inclusion of viscous dissipation, then it would be beneficial to expand upon this topic (and include the body of work from alpine glaciers) (e.g., Andrews et al., 2014; Gordon et al., 1998; Hodge, 1979; Murray and Clarke, 1995), perhaps in a separate section or paragraph. However, the manuscript should also note that a through modeling effort to explore this future work.

Page 5 4-5. It would be useful to expand on the representation of channels in this model compared to other models because they are very different - previous models represent channels along element edges (e.g., Hewitt, 2013; Schoof, 2010; Werder et al., 2013).

5-6. Is two-way coupling implemented between ShaKTI and ISSM?

21-22. Clearly define that $\beta$ is a function of bedrock bump height and spacing and that it goes to zero when the gap height exceeds the bedrock bump height. This is essentially the delineation between 'cavity type' opening and 'channel type' opening and shouldn't be relegated to the Tables alone.

26. What does $\omega$ represent, more than simply the 'Parameter controlling nonlinear transition between laminar and turbulent flow'? In order to be useful to readers, some information about how it is chosen needs to be provided.

26. What is the characteristic length scale used in the calculation of the Reynolds number? This length scale should be associated with bedrock bump spacing and the gap height though some sort of hydraulic radius. How this is characterized and justification should be discussed. In this vein, q in Table 1 should probably have an equation associated with it.

Page 6 10-29. It would be useful to mention that the internal dissipation term in Equation 10 is not included in the Werder et al. (2013) formulation and perhaps nod to previous discussion of the inclusion (or lack thereof) of this term in previous modeling

efforts (see comment on p4L13-14).

Page 7 3. Awkward phrasing.

13-15. The over/under pressure problem is complex (Hewitt et al. (2012) and Schoof et al. (2012) only solve it in one dimension). It may be best to temper this statement and simply explain why subglacial pressures are constrained and how the forces are balanced.

15. Extra ';'

Page 8 19-20. The grid scale and the duration of the model run should be mentioned.

30-31. Rather than stating that the head and gap heights show a clear channelization structure, why not plot the 'degree of channelization'? This will remove any ambiguity.

Page 9 1. consider using the term 'arborescent'.

5-8. This should be moved and expanded into a model limitations section and the supplementary figure should move to the main text.

7-8. Quite similar might be an overstatement, particularly because differences in the vicinity of channels is +30 meters - which is ∼10% of the total ice thickness and ∼50% of the total diurnal head variation measured by Andrews et al. (2014).

23-24. What low distributed input value? Does the choice of initial subglacial gap height affect the spin up time?

28. Though the meltwater input during the winter is low, it really isn't realistic. Is there a model stability reason for having winter meltwater input?

Page 10 6-8. These sentences imply that the model is fully coupled with ISSM. Unless this is the case, consider adding citations to delineate that the described ice velocity behavior is what would be expected in the coupled model, or rephrase the sentences.

16-18. The last sentence in this paragraph is a bit out of place.

Page 11 7-9. This sentence should have a citation to minimize confusion between the model results here and the link between ice velocity and the subglacial system.

9-11. It makes sense to try and relate this work to observational work on outlet glaciers since those are the glaciers most likely to impact sea level rise, but the boundary conditions and the model domain presented here are more realistic for land-terminating regions of the ice sheet.

15-20. This paragraph should be expanded into a 'model limitations' section. It would also be nice to see some discussion of an ideal length scale. I imagine that when coupled with an ice dynamical model, there will be some grid size after which, a finer mesh won't improve modeling results due to modeled ice characteristics.

Figures Figure 2. It would useful to see the 'degree of channelization'. Also consider using a non-linear color scale for gap height and flux.

Figure 3. Can the gap height panels be plotted on the same scale? It would also be useful to see the 'degree of channelization'

Figure 4. 'Box on' for panels b and c. Panel labels are also needed. Consider adding 'degree of channelization'. Instead of using the log of gap height, consider just using a nonlinear color bar (for this and all other figures).

SI figure. This figure should move to the main text and include difference plots of channelization and possibly gap height.

References Andrews, L. C., Catania, G. A., Hoffman, M. J., Gulley, J. D., Lüthi, M. P., Ryser, C., Hawley, R. L. and Neumann, T. A.: Direct observations of evolving subglacial drainage beneath the Greenland Ice Sheet, Nature, 514(7520), 80-83, doi:10.1038/nature13796, 2014. Bartholomew, I. D., Nienow, P., Mair, D., Hubbard, A., King, M. A. and Sole, A.: Seasonal evolution of subglacial drainage and acceleration in a Greenland outlet glacier, Nature Geosci, 3(6), 408-411, doi:10.1038/ngeo863, 2010. Chandler, D. M., Wadham, J. L., Lis, G. P., Cowton, T., Sole, A., Bartholomew, I.,

Telling, J., Nienow, P., Bagshaw, E. B., Mair, D., Vinen, S. and Hubbard, A.: Evolution of the subglacial drainage system beneath the Greenland Ice Sheet revealed by tracers, Nature Geosci, 6(3), 195-198, doi:10.1038/ngeo1737, 2013. Downs, J. Z., Johnson, J. V., Harper, J. T., Meierbachtol, T. and Werder, M. A.: Dynamic hydraulic conductivity reconciles mismatch between modeled and observed winter subglacial water pressure, Journal of Geophysical Research: Earth Surface, doi:10.1002/2017JF004522, 2018. Flowers, G. E.: Modelling water flow under glaciers and ice sheets, Proceedings of the Royal Society A: Mathematical, Physical and Engineering Sciences, 471(2176), 20140907-20140907, doi:10.1098/rspa.2014.0907, 2015. Gordon, S., Sharp, M., Hubbard, B., Smart, C., Ketterling, B. and Willis, I.: Seasonal reorganization of subglacial drainage inferred from measurements in boreholes, Hydrol. Process., 12(1), 105-133, doi:10.1002/(SICI)1099-1085(199801)12:1<105::AID-HYP566>3.0.CO;2-#, 1998. Hewitt, I. J.: Seasonal changes in ice sheet motion due to melt water lubrication, Earth and Planetary Science Letters, 371-372, 16-25, doi:10.1016/j.epsl.2013.04.022, 2013. Hewitt, I. J., Schoof, C. and Werder, M. A.: Flotation and free surface flow in a model for subglacial drainage. Part 2. Channel flow, Journal of Fluid Mechanics, 702, 157-187, doi:10.1017/jfm.2012.166, 2012. Hodge, S. M.: Direct Measurements of Basal Water Pressures: Progress and Problems, Journal of Glaciology, 23(89), 309-319, 1979. Hoffman, M. J. and Price, S.: Feedbacks between coupled subglacial hydrology and glacier dynamics, J. Geophys. Res. Earth Surf., 119(3), 414-436, doi:10.1002/2013JF002943, 2014. Kamb, B.: Glacier surge mechanism based on linked cavity configuration of the basal water conduit system, Journal of Geophysical Research B, 92(B9), 9083-9100, 1987. Murray, T. and Clarke, G. K. C.: Black-box modeling of the subglacial water system, J. Geophys. Res., 100(B6), 10231-10245, doi:10.1029/95JB00671, 1995. Rada, C. and Schoof, C.: Subglacial drainage characterization from eight years of continuous borehole data on a small glacier in the Yukon Territory, Canada, The Cryosphere Discuss., 2018, 1-42, doi:10.5194/tc-2017-270, 2018. Schoof, C.: Ice-sheet acceleration driven by melt supply variability, Nature, 468(7325), 803-806, doi:10.1038/nature09618, 2010. Schoof, C., Hewitt,
I. J. and Werder, M. A.: Flotation and free surface flow in a model for subglacial drainage. Part 1. Distributed drainage, Journal of Fluid Mechanics, 702, 126-156, doi:10.1017/jfm.2012.165, 2012. Shannon, S. R., Payne, A. J., Bartholomew, I. D., Broeke, M. R. van den, Edwards, T. L., Fettweis, X., Gagliardini, O., Gillet-Chaulet, F., Goelzer, H., Hoffman, M. J., Huybrechts, P., Mair, D. W. F., Nienow, P. W., Perego, M., Price, S. F., Smeets, C. J. P. P., Sole, A. J., Wal, R. S. W. van de and Zwinger, T.: Enhanced basal lubrication and the contribution of the Greenland ice sheet to future sea-level rise, PNAS, 110(35), 14156-14161, doi:10.1073/pnas.1212647110, 2013. Walder, J. S.: Hydraulics of subglacial cavities, Journal of Glaciology, 32(112), 439-445, 1986. Werder, M. A., Hewitt, I. J., Schoof, C. G. and Flowers, G. E.: Modeling channelized and distributed subglacial drainage in two dimensions, Journal of Geophysical Research: Earth Surface, 118(4), 2140-2158, doi:10.1002/jgrf.20146, 2013.

---

## Referee Comment (RC2) · Anonymous Referee #2 · 27 Apr 2018

Summary

In the context of the current proliferation of subglacial hydrology models that largely replicate one another or make incremental changes to existing models, this paper outlines a contribution that seems a potentially worthy advancement. The model formulation is uniform across the domain and permits all the physical processes normally invoked within the spectrum of "distributed" to "channelized" drainage, with a smooth transition between laminar and turbulent flow regimes. This formulation is conceptually simple and allows fast and slow drainage regimes to evolve in an organic and intuitive fashion. The authors have incorporated the model into an established ice-flow model, making it in principal readily available to the wider community.

General comments

[Figure]

As elaborated below, I think the description of this new model has the potential to make a strong contribution to GMD if the authors consider the following revisions (roughly in order of importance):

(1) Adding technical model detail commensurate with (assumed) expectations for a journal focused on model development, including a more thorough elaboration of model boundary conditions, implementation and numerics; (2) Amplifying the description of the conceptual model and more thoroughly justifying the choices made in model formulation; (3) Reporting on the results of basic model testing: model convergence, consistency, efficiency, grid refinement (done to an extent already), etc. and presenting quantitative evidence of model performance (e.g. runtimes); (4) Addressing issues that plague many models of subglacial hydrology and being up-front about the shortcomings of the current model (or better showcasing the successes). Examples of these issues are: (a) low winter water pressures in contrast to observations, (b) englacial storage motivated by numerical need, (c) extreme sensitivity to initial and boundary conditions, (d) maintaining saturated conditions, (e) water-mass conservation when pressures are capped at overburden (f) convergence in the presence of substantial bed topography (g) fundamental continuum assumptions and omission of the unconnected bed (h) prescription of constant sliding speed and omission of two-way coupling (5) Streamlining the introductory material and omitting or condensing content that anyone reading this paper with the intention of using the model should already know very well; (6) Dialing back some of the stated advances of SHaKTI over existing models;

I would consider some amount of revision in response to items (1)-(4) to be essential (see details below), and revisions in response to (5) and (6) desireable.

Detailed comments (page.line)

SHaKTI: Not sure exactly what "kinetic transient interactions" are and why this phrase forms an essential part of the model name. "kinetic transient interactions" sounds more like a biochemistry term. It would help the readers if the authors could use the

full model name in a sentence to make it clear why this acronym was chosen, aside from its perhaps appealing phonetic similarity to "chakra".

1.5 "changes the governing physics under different flow regimes" If this were a clunky IF-THEN sort of statement in standard models, then I see the point. Models like those of Hewitt (2013) and Werder et al (2013) have all the governing physics but simply apply the appropriate governing equations to different parts of the model mesh (edges versus cells). Perhaps the point to emphasize here is that SHaKTI, in principal, may capture intermediate flow regimes with the laminar-turbulent transition. One could argue, however, that the other models also do this by having channels and cavity systems operating simultaneously and in spatial proximity, thus together forming intermediate flow regimes.

1.13-14 "supporting the notion that..." delete. Too obvious.

1-2. Suggest condensing introduction and omitting textbook-level content, e.g., lines 16-18.

2.28-3.10. Suggest omitting or highly condensing this very basic background material. See Flowers (Proceedings of the Royal Society A, 2015) for a convenient citation to replace much of this content. Ditto for most of page 3.

4.16. I'm not sure most glaciologists would agree that using different governing equations for fundamentally distributed and channelized drainage systems is questionable. Perhaps emphasize the lack of intermediate flow regimes as in the next sentence. Here I think the drawback of existing models are overstated.

4.26. Replace "it is satisfying"

4.28. Not clear how this model allows "high-resolution" exploration in particular.

5.1-8. Please give an overview of the conceptual model here. How is the drainage element envisioned? How does this relate to the fracture-flow formulation of q?

5.10-11. Conservation of water AND ice mass? "basal water flux" => "horizontal water flux"; define "internal melt generation". Not clear if that would be englacial melt that makes its way to the basal drainage system or something else.

5.12-13. I struggle to see how SHaKTI "can be viewed as an approximation to a multidimensional generalization of the governing equations for glacial conduits described by Spring and Hutter (1981) and Clarke (2003)." These references describe only channel physics, not opening by sliding as in cavities. Clarke uses conduit distensibility in the governing equations and accounts for thermal advection, in contrast to SHaKTI. Easiest just to omit this text. I don't think trying to explain the statement would add much.

5.15. It looks like the theory is developed for fully saturated flow, so this should be stated explicitly.

5.18. "input rate" = "internal melt rate" above?

5.20 (Eqn 2). Better described as "evolution of gap height" than "gap dynamics"? State what these terms are before the end of the paragraph, ideally before the equation. Eqn (2) would appear to allow for creep opening, not just creep closure. Is this intentional? If not, why write the creep term in this way? Is creep opening permitted in the numerical implementation of the model? If so, it should be justified.

5.26 (Eqn 3). State that this is the formulation for fracture flow, or how this formulation came to be adopted. Define Re here as in table, else the laminar-turbulent transition doesn't make sense. Here the reader really needs to know what the conceptual model is in order to make sense of the flux formulation.

6. This reader is wondering how b and h are going to be related in the model, as the treatment of gap height and water pressure/hydraulic head forms a key difference in various models. Perhaps mention this early on when saturation conditions are noted.

6.6. fracture flow: this is a description of the conceptual model that should appear

earlier.

6.7 "Most" => "Many"

6.14-15. "heat consumed due to changes in water pressure" More physically based to explain that it is the heat consumed or released in maintaining the water at the pressure-melting temperature in the presence of changing water pressure.

6.17-18. Good place to cite Clarke (2003) for heat advection and Creyts and Clarke (2010) for supercooling.

6.21. Please state rationale for including englacial storage. Werder et al (2013) do this, but is it needed here for numerical stability?

6.28-30. Expressing K as a tensor here, given that it is assumed isotropic, seems needlessly complicated. An even more compact way to write the first term in Eqn (9) is \nable \dot q.

7. Section 2.2. Boundary conditions are key for model implementation. It would seem to make sense to articulate them mathematically. I think Werder et al (2013) set a nice example of the balance between the mathematical and descriptive exposition of a model, including boundary conditions and method of solution.

7.9. So, negative water pressures are permitted in the model? If so, how big are they? Do they have a significant influence on creep closure?

7.12. How is the Pw=Pi restriction implemented without violating conservation of mass? If it's not, it would be good to report the amount of mass-conservation violation this restriction imposes.

7.17. "Euler-Backward" => "backward Euler" seems more conventional, unless this means something else.

7.17. Picard iteration. This is a common methodology, but one not known for its speed. Though not mentioned in the manuscript, I surmise that a major advantage of this

**[GMDD](GMDD)**
modeling approach (unified physics applied everywhere) over others could be its efficiency, but perhaps not, depending on the numerical implementation. This reader would be very interested to known if the model formulation had the potential to be fast, and whether the numerical implementation was designed with this in mind. Given the disparity in timescales between ice flow and water flow which typically necessitates comparatively small timesteps for hydrological models, many model users will be looking for hydrological models that do not add unnecessary computational burden to their ice-sheet models.

7.19-20. It sounds like gap height and hydraulic head are not solved simultaneously (or iteratively). Why not? Explicit time-stepping is simple but can lead to large errors. Can the authors reassure the readers that this has been investigated and propose corresponding limits on the time step?

8.1. Is the model convergence sensitive to prescribed initial gap height?

8.7. Curious why closure is not included in "degree of channelization". If closure balances opening at small gap heights for any opening mechanism, it seems channelization would be supressed.

8.11-12. The software-style description seems a little strange. I guess the key thing here is that the output is ascii, not binary or something else? It seems like output from any model could be visualized in contour plots, timeseries, etc, and in any software.

8.15. Somewhere above this it should be noted that the mesh is irregular.

8.16. "Application". Here I was expecting to see some multi-faceted demonstration of the model performance (e.g. accuracy, consistency, convergence, efficiency) prior to the demonstration that the model produces qualitatively familiar results in some basic tests. Model performance metrics are not often reported in journal articles focused on model applications, but I expected this would be different in Geoscientific Model Development. The Editor can decide if this suggestion is misguided; it could be misinterpreting the purpose and expectations of the journal.

8-9. Subglacial hydrology models frequently have trouble in the presence of bed topography. The tests presented here omit bed topography with the exception of a gentle slope. It would be useful to know if this model does better than others in the presence of realistic bed topography. It's ok if it doesn't.

9.3. "drainage configuration ... affected by ... bed topography" Except in this test the bed is flat. Is this just a general statement?

9.4 "unstructured mesh" Please mention this when model implementation is described.

9.10. The test domains seem very small. It would be useful to report something on model runtimes. Is it practical to run this model coupled to an iceflow model for a large catchment?

10.3. "do not include storage term" Meaning englacial storage?

10.6-7. Paper should make clear that ub is prescribed and constant, thus there is no two-way coupling with sliding, meaning the negative feedback associated with sliding is absent (Hoffman and Price, 2014?)

10.9. This sounds like a problem that plagues most models (c.f. Downs et al, 2018: https://agupubs.onlinelibrary.wiley.com/doi/full/10.1002/2017JF004522), so should be noted as a common shortcoming with a citation or two.

10.21/ "impose potential channel locations" Indeed this is a limitation in some models, but here the "channel" locations are a function of the mesh, just as they are in the models of Hewitt, Schoof and Werder. In the latter case, the channels may lie anywhere along the mesh edge. In this model, they may lie anywhere in the mesh elements. In the models of Hewitt, Schoof, Werder, increases in grid size mean small channels cannot be represented; in the current model, increases in grid size mean channels become unrealistically wide. Both are limitations in different ways.

[Figure]

10.25-26. This seems like a big deal, and a true potential advantage over the other models out there.

11.10-11. "Supports the notion" Too obvious. Suggest deleting.

11.13-14. Arguably the unconnected bed requires additional model physics, but this regime has been parameterized by Hoffman et al (2016) and Downs et al (2018).

11.25 suggest "channels" => "pathways". Reword "sorts itself out".

Editorial (page.line)

2.22. "the model..., a model formulation" => "we describe the model formulation of SHaKTI, which allows for..."

5.19 and 5.25. These lines and the text that follows them do not form sentences.

6.9. ditto above

References: more than 15 of the references are incomplete. Authors should check the list thoroughly. "Truffer" is missing an "r". "et al" is used where it probably shouldn't be. Sometimes journal titles are written out, sometimes they are not.

Tables: check superscripts. I have great respect for SI, but please give ub in m/a also.

---

## Short Comment (SC1) · 27 Apr 2018

Link https://issm.jpl.nasa.gov/download/ given in the manuscript to access the code is not working. This should be fixed.

Lutz Gross GMD Executive Editor

---

## Short Comment (SC2) · 1 May 2018

The link to access the code as given in the manuscript (https://issm.jpl.nasa.gov/download/) should be working. It seems that JPL was having a network issue last week, but it is now fixed.

Thanks, Aleah Sommers

---

## Author Comment (AC1) · 18 Jun 2018

**Response to Reviewer 1**

We thank the reviewer for providing a supportive and constructive review of this work. Your insightful suggestions and comments have substantially improved the clarity of the revised manuscript. Below are responses addressing each comment. A revised manuscript is included following our responses to both reviewers.

Sincerely,

Aleah Sommers

RC1 comments (in blue):

General comments

1. The introduction should be more clearly focused (and quite possibly expanded) on the topic of subglacial hydrology. There is a fairly extensive body of literature about subglacial model development, including extensive work on alpine glaciers. The focus on outlet glaciers and sea level rise in the introduction is somewhat of an aside.

We have revised the introduction (Section 1) to be more general, but we feel that the implication of increased mass loss from glaciers and ice sheets as contributors to sea level rise provides broader context and is the primary big-picture motivation for this work and the increasing body of research on the response of ice sheets to climate change. For this reason, we retain some emphasis on outlet glaciers and sea level rise.

2. The motivation of the manuscript is somewhat unclear if the reader is un- indoctrinated into the world of subglacial hydrology. It would be useful to include a through description of viscous dissipation and why it hasn't been included in previous subglacial models in section 1.2 and clearly describe - before the model description - the goals of this modeling effort.

We have revised Sections 1.1 and 1.2 to include a more clear explanation of mechanical energy dissipation and the problems that have arisen with including this term in other formulations, as well as a clear statement of the goal to see if we could use a single set of governing equations to produce systematic self-organized channelization where it should occur.

3. The basal flux parameterization (Line 25) needs to be more carefully documented. There are several line notes to this effect, but essentially, the addition of the Reynolds number requires the selection of characteristic length scales and dimensionless parameters - reasoning behind how these values are assigned should be included in order to enhance the usefulness of this manuscript.

Section 2.1 in the revised manuscript now includes an elaboration of the flux formulation

(Eq. 5 in the revised manuscript), the different terms involved, and a description of its basis in fracture flow and derivation.

Page 1 Line 1. I am not sure "poorly understood" is the best phrase to use here. There is an extensive body of literature exploring the state and evolution of the subglacial hydrologic system and its representation in current models, while not perfect, are able to replicate many features of ice velocity fields. We know that the link between melt and ice motion is the subglacial system; however, there are parameters and parameterizations which are not well constrained.

This wording has been changed to "not fully understood".

1-2. The wording of this sentence is awkward.

This wording has been revised.

9. Much of the manuscript switches between 'channel' and 'efficient' drainage. Con- sider using something like '. . .over a wide range of drainage efficiencies. . .' or inefficient and efficient drainage to eliminate the "channel".

After careful consideration, we feel that using "channelized" and "sheetlike" drainage is more descriptive than "efficient" and "inefficient" drainage, and helps the reader better visualize the drainage configuration.  In the revised manuscript, we have made an effort to be consistent in our use of the terms, while clarifying early in the paper that channelized drainage refers to efficient drainage, and sheetlike drainage refers to inefficient or distributed drainage.

15-22. While understanding ice sheet dynamics is important for the characterization of future sea level rise, it might be more correct to acknowledge that basal lubrication alone may not be a major uncertainty in sea level rise predictions (e.g., IPCC, sea level change chapter, pages 1168-1169; Shannon et al., 2013).

This text has been updated (lines 17-18 of revised manuscript).

21-22. Consider citing the chapter instead of the whole 'Physical Basis' document.

Citation changed (line 18): Church, J.A., P.U. Clark, A. Cazenave, J.M. Gregory, S. Jevrejeva, A. Levermann, M.A. Merrifield, G.A. Milne, R.S. Nerem, P.D. Nunn, A.J. Payne, W.T. Pfeffer, D. Stammer and A.S. Unnikrishnan, 2013: Sea Level Change. In: Climate Change 2013: The Physical Science Basis. Contribution of Working Group I to the Fifth Assessment Report of the Intergovernmental Panel on Climate Change [Stocker, T.F., D. Qin, G.-K. Plattner, M. Tignor, S.K. Allen, J. Boschung, A. Nauels, Y. Xia, V. Bex and P.M. Midgley (eds.)]. Cambridge University Press, Cambridge, United Kingdom and New York, NY, USA.

7. There are a number of other publications that could be cited along with Cowton et al. (2013), including Bartholomew et al. (2010), Chandler et al. (2013), and Andrews et al. (2014).

Citations added (Page 2, lines 9-10).

8-9. This sentence seems out of place and doesn't provide much information.

This sentence was removed.

12-13. It seems that a description of the unknowns would logically follow this sentence instead of a description of a description of how the subglacial system works. It might be useful to remove references to unknowns - while this is certainly true - the main purpose of the manuscript is to rectify a persistent known problem - that models of subglacial hydrology tend to only represent 2 endmembers of the continuum of possible configurations.

This sentence was removed.

23-26. Reference the section numbers

Changes have been made in the revised manuscript to reference the section numbers (page 2, line 33 to page 3, line 2).

27. This section would benefit from explaining the motivation for subglacial hydrology model development as well. See Flowers (2015) for a great review of the topic.

It was a major oversight to neglect citing Flowers (2015) in this section. We are familiar with the review and have greatly benefited from it; we thank the reviewer for pointing out the omission. In the revised manuscript, we clearly point toward that paper as a resource (Section 1.1).

3-4. This relates to the previous comment - it would be nice to detangle why the community ended up focusing on these two endmembers.

We have attempted to clarify this in the revised manuscript (Sections 1.1-1.2).

13. See comments regarding line p2L12-13 and p1L1.

Text changed: "Although the effects of surface melt on ice sheet dynamics are not yet entirely understood…" (page 3, line 21).

34 - p4L5. These sentences start to feel rushed. Also consider including more recent work by Rada and Schoof (2018) and Downs et al. (2018).

The text has been revised (page 4, lines 8-16).

Page 4 7-8. This sentence is a direct repeat of a sentence in the abstract. Consider revising.

This text has been revised (page 4, lines 18-19).

13-15. The instability that arises with the viscous dissipation has been discussed by a number of studies (Hewitt et al., 2012; Hoffman and Price, 2014; Kamb, 1987; Schoof, 2010; Schoof et al., 2012; Walder, 1986; Werder et al., 2013). In addition, Flowers (2015) has a nice summary of the reasoning behind and numerical approaches to switching between drainage elements. Because the primary contribution of this work is the inclusion of the viscous dissipation term and the representation of both turbulent and laminar flow, it is important to thoroughly discuss the reasoning and justification and numerics used in previous modeling work. This summary could readily follow lines 13-15.

We have attempted to address this by including a clearer description and explanation in the revised manuscript (Section 1.2).

18. If the isolated/weakly connected system is the primary scientific motivation behind the inclusion of viscous dissipation, then it would be beneficial to expand upon this topic (and include the body of work from alpine glaciers) (e.g., Andrews et al., 2014; Gordon et al., 1998; Hodge, 1979; Murray and Clarke, 1995), perhaps in a separate section or paragraph. However, the manuscript should also note that a through modeling effort to explore this future work.

The isolated/weakly connected system is not the main motivation for including the dissipation term, but it is a challenge that faces many existing subglacial hydrology models. We do not specifically attempt to produce unconnected regions, but the flexible configuration of the geometry in our model may be conducive for allowing these regions to exist with appropriate topography.

Page 5 4-5. It would be useful to expand on the representation of channels in this model compared to other models because they are very different - previous models represent channels along element edges (e.g., Hewitt, 2013; Schoof, 2010; Werder et al., 2013).

The text has been revised (Section 1.2).

5-6. Is two-way coupling implemented between ShaKTI and ISSM?

Not yet, although that is planned for upcoming work.  We have clarified this in the revised text (page 5, lines 24-26; also see Section 4.1: page 15, lines 24-28).

21-22. Clearly define that $\beta$ is a function of bedrock bump height and spacing and that it goes to zero when the gap height exceeds the bedrock bump height. This is essentially the delineation between 'cavity type' opening and 'channel type' opening and shouldn't be relegated to the Tables alone.

The text has been updated and $\beta$ clearly defined (page 6, lines 23-25, Eqs. 3 and 4).

26. What does $\omega$ represent, more than simply the 'Parameter controlling nonlinear transition between laminar and turbulent flow'? In order to be useful to readers, some information about how it is chosen needs to be provided.

$1/\omega$ represents a Reynolds number at which a departure from laminar flow behavior becomes significant, and the square-root turbulent dependence becomes dominant. We have elaborated in the revised text to include a more detailed description and explanation of the flux formulation in the revised manuscript and the role of $\omega$ (page 7).

26. What is the characteristic length scale used in the calculation of the Reynolds number? This length scale should be associated with bedrock bump spacing and the gap height though some sort of hydraulic radius. How this is characterized and justification should be discussed. In this vein, q in Table 1 should probably have an equation associated with it.

The Reynolds number depends on the flux q, using the gap height as a characteristic length. This has been explicitly added to the revised manuscript (see Eq. 7). Additional explanation is included in Section 2.1 following Eq. (5) on page 7 to clarify the basis and derivation of our flux formulation. The equation for Re has also been added to Table 1 for clarity.

Page 6 10-29. It would be useful to mention that the internal dissipation term in Equation 10 is not included in the Werder et al. (2013) formulation and perhaps nod to previous discussion of the inclusion (or lack thereof) of this term in previous modeling

The text has been revised (page 8, lines 15-17).

This text has been revised (page 9, lines 4-5).

13-15. The over/under pressure problem is complex (Hewitt et al. (2012) and Schoof et al. (2012) only solve it in one dimension). It may be best to temper this statement and simply explain why subglacial pressures are constrained and how the forces are balanced.

The wording has been changed in the revised manuscript (page 9, lines 19-21).

Removed in updated manuscript.

Page 8 19-20. The grid scale and the duration of the model run should be mentioned.

These details have been included in the revised manuscript (Section 3.1).

30-31. Rather than stating that the head and gap heights show a clear channelization structure, why not plot the 'degree of channelization'? This will remove any ambiguity.

Degree of channelization is included in the revised figures.

This is a word with nice imagery and historical context in subglacial hydrology; it has been included in the revised manuscript (page 11, line 12).

5-8. This should be moved and expanded into a model limitations section and the supplementary figure should move to the main text.

The supplementary figure has been revised and is now included in the main text as Fig. 4. We keep the discussion of mesh dependence for this example here where it first appears in the simulation results (Section 3.2) and revisit the topic in the new "Model Limitations" section 4.1.

7-8. Quite similar might be an overstatement, particularly because differences in the vicinity of channels is +30 meters - which is ~10% of the total ice thickness and ~50% of the total diurnal head variation measured by Andrews et al. (2014).

This wording has been changed and the figures revised. In the revised manuscript, the meshes were adjusted to ensure that each moulin input was truly located at the same location, and the results more clearly illustrate mesh convergence and the local variations that arise in regions of channelization (Figs. 3 and 4).

23-24. What low distributed input value? Does the choice of initial subglacial gap height affect the spin up time?

1 m a$^{-1}$ distributed input (updated in text; page 12, lines 5-13). We acknowledge this is unrealistically high, as are the magnitudes of distributed melt input in our transient example; these extreme values are used to demonstrate the stable transition to self-organized channelization with very high forcing.

No, the initial gap height does not significantly affect the spin-up time for initial gap height of 0.001 to 0.1 m; this is now stated in Section 4.1 (Model Limitations).

28. Though the meltwater input during the winter is low, it really isn't realistic. Is there a model stability reason for having winter meltwater input?

Yes. With zero meltwater input everywhere, the system tries to shut itself down and the nonlinear iteration has trouble converging. The model does perform well with zero distributed meltwater input if there are other point inputs somewhere in the domain (like in the single moulin and 10 moulin examples in this paper, Sections 3.1-3.2). In reality, englacial discharge may be lagged with some low input to the subglacial system occurring through winter.

Page 10 6-8. These sentences imply that the model is fully coupled with ISSM. Unless this is the case, consider adding citations to delineate that the described ice velocity behavior is what would be expected in the coupled model, or rephrase the sentences.

We hold velocity constant, but describe the pressure changes and how that would relate to sliding velocity in a coupled model. The text has been updated in the revised manuscript to avoid misleading wording (page 12, lines 23-27).

16-18. The last sentence in this paragraph is a bit out of place.

This sentence has been moved to Section 1.2.

Page 11 7-9. This sentence should have a citation to minimize confusion between the model results here and the link between ice velocity and the subglacial system.

The wording has been changed in the revised manuscript.

9-11. It makes sense to try and relate this work to observational work on outlet glaciers since those are the glaciers most likely to impact sea level rise, but the boundary conditions and the model domain presented here are more realistic for land-terminating regions of the ice sheet.

The text has been updated (page 14, lines 11-13). Our model can be applied to either land-terminating or marine-terminating glaciers, with different boundary conditions. We have tested the model on a marine-terminating glacier with successful results (forthcoming, not included in this paper).

15-20. This paragraph should be expanded into a 'model limitations' section. It would also be nice to see some discussion of an ideal length scale. I imagine that when coupled with an ice dynamical model, there will be some grid size after which, a finer mesh won't improve modeling results due to modeled ice characteristics.

A Model Limitations subsection has been added in the revised manuscript (Section 4.1) that includes discussion of an ideal length scale. Also note the additional exploration of mesh dependence included in the revised manuscript for the examples in Sections 3.2 and 3.3.

Figures

Figure 2. It would useful to see the 'degree of channelization'. Also consider using a non-linear color scale for gap height and flux.

This figure has been revised. (Fig. 2)

Figure 3. Can the gap height panels be plotted on the same scale? It would also be useful to see the 'degree of channelization'

This figure has been revised. (Fig. 3)

Figure 4. 'Box on' for panels b and c. Panel labels are also needed. Consider adding 'degree of channelization'. Instead of using the log of gap height, consider just using a nonlinear color bar (for this and all other figures).

We have revised the figures to include degree of channelization and now use log-scale color bars for gap height, flux, and degree of channelization to show detailed structure rather than plotting the log of quantities.

SI figure. This figure should move to the main text and include difference plots of channelization and possibly gap height.

This figure has been revised and moved to the main text as Fig. 4. Due to the slightly offset locations of specific channels in unstructured meshes, difference plots may in some cases be misleading by indicating a large error, which is in reality a quite similar value but slightly offset due to mesh variations. To more clearly illustrate mesh dependence/convergence and areas of local variation, we have revised Figs. 3 and 4 to include a broader range of mesh resolution and compare y-averaged pressure distributions.

**Response to Reviewer 2:**

Thank you for your detailed and helpful review of this manuscript. We have attempted to address the concerns and incorporate your recommendations into a revised version of the manuscript. Below we respond to individual comments. A revised manuscript is included following the responses.

Sincerely,

Aleah Sommers

RC2 comments (in purple):

As elaborated below, I think the description of this new model has the potential to make a strong contribution to GMD if the authors consider the following revisions (roughly in order of importance):

(1) Adding technical model detail commensurate with (assumed) expectations for a journal focused on model development, including a more thorough elaboration of model boundary conditions, implementation and numerics; (2) Amplifying the description of the conceptual model and more thoroughly justifying the choices made in model formulation; (3) Reporting on the results of basic model testing: model convergence, consistency, efficiency, grid refinement (done to an extent already), etc. and presenting quantitative evidence of model performance (e.g. runtimes); (4) Addressing issues that plague many models of subglacial hydrology and being up-front about the shortcomings of the current model (or better showcasing the successes). Examples of these issues are: (a) low winter water pressures in contrast to observations, (b) englacial storage motivated by numerical need, (c) extreme sensitivity to initial and boundary conditions, (d) maintaining saturated conditions, (e) water-mass conservation when pressures are capped at overburden (f) convergence in the presence of substantial bed topography (g) fundamental continuum assumptions and omission of the unconnected bed (h) prescription of constant sliding speed and omission of two-way coupling (5) Streamlining the introductory material and omitting or condensing content that anyone reading this paper with the intention of using the model should already know very well; (6) Dialing back some of the stated advances of SHaKTI over existing models;

Detailed comments (page.line)

SHaKTI: Not sure exactly what "kinetic transient interactions" are and why this phrase forms an essential part of the model name. "kinetic transient interactions" sounds more like a biochemistry term. It would help the readers if the authors could use the full model name in a sentence to make it clear why this acronym was chosen, aside from its perhaps appealing phonetic similarity to "chakra".

The name SHAKTI is intended to highlight the complex interactions through movement of

water and ice (the term "kinetic" refers to motion in this context), and the fact that the subglacial system evolves with time ("transient"). A description has been added for clarity in Section 1 when the model name is first presented (page 2, lines 26-27), as well as a comma in the name after the word "kinetic" to help avoid confusion with chemical kinetics.

Note: "Shakti" is a Sanskrit term for energy, which gives form to everything in the universe. This could be seemingly unrelated to subglacial hydrology in particular, but we contend that it is highly relevant to all physical phenomena. In fact, the energy dissipated by flowing water plays an important role in generating the subglacial hydrologic system, as we see by including the dissipation term in our model formulation.

1.5 "changes the governing physics under different flow regimes" If this were a clunky IF-THEN sort of statement in standard models, then I see the point. Models like those of Hewitt (2013) and Werder et al (2013) have all the governing physics but simply apply the appropriate governing equations to different parts of the model mesh (edges versus cells). Perhaps the point to emphasize here is that SHaKTI, in principal, may capture intermediate flow regimes with the laminar-turbulent transition. One could argue, however, that the other models also do this by having channels and cavity systems operating simultaneously and in spatial proximity, thus together forming intermediate flow regimes.

The text has been revised in the updated manuscript (page 1, lines 4-6): "Imposing a distinction that applies different equations to capture the dominant physics in different parts of the model domain, however, may not allow for the full array of drainage characteristics to arise".

1.13-14 "supporting the notion that. . ." delete. Too obvious.

This text has been deleted.

1-2. Suggest condensing introduction and omitting textbook-level content, e.g., lines

16-18.

The Introduction (Section 1) has been revised while attempting to maintain sufficient background explanation for anyone not familiar with subglacial hydrology (pages 1-4). Reviewer 1 of this paper suggested that the introduction should be broader, so we have aimed to strike a balance between the recommendations of both reviewers. We hope that the revised introduction helps to better place this work in context of previous work in subglacial hydrology and the overall context of ice sheet response to climate change.

2.28-3.10. Suggest omitting or highly condensing this very basic background material. See Flowers (Proceedings of the Royal Society A, 2015) for a convenient citation to replace much of this content. Ditto for most of page 3.

We have revised Section 1.1 and included a clear reference to Flowers (2015) at the start. Although this paper is focused on the model formulation, we feel it is necessary to include an ample description of previous work to place our model in context of what has come

before. This follows the same line of reasoning as our response to the previous comment.

4.16. I'm not sure most glaciologists would agree that using different governing equations for fundamentally distributed and channelized drainage systems is questionable. Perhaps emphasize the lack of intermediate flow regimes as in the next sentence. Here I think the drawbacks of existing models are overstated.

As pointed out by the reviewer, there is indeed a continuum of flow morphologies that develop in subglacial environments that include intermediate flow regimes. From a fundamental perspective, there is no need to use different governing equations for distributed versus channelized drainage systems – each system requires statements of water mass conservation, ice mass conservation, momentum balance and energy balance, including relevant terms. Parameterizations of various terms in the governing equations may vary – for example, the creep closure term can be parameterized for channels by invoking the approximately semi-circular geometry, whereas a different parameterization is used for distributed systems where creep closure occurs between supporting bedrock bumps. We do not disagree that different parameterizations of various terms in the governing equations may be suitable for distributed versus channelized drainage. However, the governing equations should be similar. More specifically, we meant to emphasize the inclusion or not of the dissipation term in the energy equation in the distributed drainage system here. To avoid misunderstanding, we have revised the text in Section 1.2.

4.26. Replace "it is satisfying"

This text has been revised.

4.28. Not clear how this model allows "high-resolution" exploration in particular.

Because the model would not capture effects of channelization on a very large grid, it is not necessarily appropriate for very coarse resolution studies (elements spanning several km or larger), which is what we intended to convey. We have added additional discussion relating to mesh dependence to the revised manuscript in Sections 3.2, 3.3, and 4.1. We agree with the reviewer that the term "high-resolution" does not contribute much meaning, and the text has been revised.

5.1-8. Please give an overview of the conceptual model here. How is the drainage element envisioned? How does this relate to the fracture-flow formulation of q?

The subglacial drainage system is represented as a sheet with variable gap height. This is now more clearly stated in the revised text (page 5, lines 18-19).

5.10-11. Conservation of water AND ice mass? "basal water flux" => "horizontal water flux"; define "internal melt generation". Not clear if that would be englacial melt that makes its way to the basal drainage system or something else.

The text has been revised for clarity (page 5, line 30 to page 6, line 1).

5.12-13. I struggle to see how SHaKTI "can be viewed as an approximation to a multi-dimensional generalization of the governing equations for glacial conduits described by Spring and Hutter (1981) and Clarke (2003)." These references describe only channel physics, not opening by sliding as in cavities. Clarke uses conduit distensibility in the governing equations and accounts for thermal advection, in contrast to SHaKTI. Easiest just to omit this text. I don't think trying to explain the statement would add much.

The most general form of the conservation equations for subglacial hydrology would be an extension of the Spring-Hutter (or Spring-Hutter-Clarke) equations to two dimensions, with augmentation to account for opening by sliding. In fact, the conduit deformation term referred to by the reviewer is analogous to the creep closure term in subglacial hydrology models. In general, a complete set of governing equations for subglacial hydrology models should include acceleration terms in the momentum equation, and advection and in-plane conduction terms should be included in the energy equation, as in the Spring-Hutter equations. This is what we were referring to. Subglacial hydrology models typically neglect the acceleration terms in the momentum equation and employ an approximate energy equation wherein all dissipated mechanical energy is locally used to produce melt (furthermore, as noted above, some models neglect internal melt generation in the sheet). We have rewritten the text for improved clarity to acknowledge the complete form of the equations and the approximations that are made here and in other models (page 6, lines 3-9).

5.15. It looks like the theory is developed for fully saturated flow, so this should be stated explicitly.

The text has been revised (page 6, lines 13-14).

5.18. "input rate" = "internal melt rate" above?

No, $i_{e \rightarrow b}$ is the external meltwater input rate (to represent surface water making its way to the bed). This has been clarified in the text (page 6, line 13). The internal melt rate is described in Eq. (11) of the revised manuscript and calculated using an energy balance at the bed.

5.20 (Eqn 2). Better described as "evolution of gap height" than "gap dynamics"? State what these terms are before the end of the paragraph, ideally before the equation. Eqn (2) would appear to allow for creep opening, not just creep closure. Is this intentional? If not, why write the creep term in this way? Is creep opening permitted in the numerical implementation of the model? If so, it should be justified.

The text has been revised to better explain this equation, and "gap dynamics" changed to "evolution of gap height" as suggested in Eq. 2 (page 6, lines 15-27).

Creep opening is not allowed, as we currently limit the water pressure to not exceed ice overburden pressure (Section 2.2).

5.26 (Eqn 3). State that this is the formulation for fracture flow, or how this formulation

came to be adopted. Define Re here as in table, else the laminar-turbulent transition doesn't make sense. Here the reader really needs to know what the conceptual model is in order to make sense of the flux formulation.

We have revised the text to include substantial elaboration following Eq. (5) to more clearly describe the basis and derivation of our flux formulation, including the Reynolds number definition and role of the transition between laminar and turbulent regimes (page 7).

6. This reader is wondering how b and h are going to be related in the model, as the treatment of gap height and water pressure/hydraulic head forms a key difference in various models. Perhaps mention this early on when saturation conditions are noted.

We obtain h by solving a nonlinear elliptic equation (or parabolic if storage is used), in which b controls the conductivity (see Eq. 15).

6.6. fracture flow: this is a description of the conceptual model that should appear earlier.

Text has been added for clarity in the opening paragraph of Section 2 (page 5, lines 18-19).

6.7 "Most" => "Many"

This wording has been changed.

6.14-15. "heat consumed due to changes in water pressure" More physically based to explain that it is the heat consumed or released in maintaining the water at the pressure-melting temperature in the presence of changing water pressure.

This wording has been revised (page 8, lines 9-10).

6.17-18. Good place to cite Clarke (2003) for heat advection and Creyts and Clarke (2010) for supercooling.

Citations have been added to the revised manuscript (page 8, line 14).

6.21. Please state rationale for including englacial storage. Werder et al (2013) do this, but is it needed here for numerical stability?

No, englacial storage is not needed for stability but is included for completeness in being able to simulate situations that may involve substantial storage.  The simulations presented in this paper use $e_v$=0 (no storage).  This has been explained in revised text (page 8, lines 18-22).

6.28-30. Expressing K as a tensor here, given that it is assumed isotropic, seems needlessly complicated. An even more compact way to write the first term in Eqn (9) is \nable \dot q.

In writing the original manuscript, we considered whether to define K as a tensor or not, and decided that it was most complete to do so, since K could potentially be anisotropic and

easily made so in the model.

In Eq. (15) of the revised manuscript (previously Eq. 9 in the original submitted manuscript), we write the first term in this form to make the dependence on h obvious to the reader.

7. Section 2.2. Boundary conditions are key for model implementation. It would seem to make sense to articulate them mathematically. I think Werder et al (2013) set a nice example of the balance between the mathematical and descriptive exposition of a model, including boundary conditions and method of solution.

The text has been updated to more clearly define boundary conditions (Section 2.2).

7.9. So, negative water pressures are permitted in the model? If so, how big are they? Do they have a significant influence on creep closure?

Negative water pressures can be calculated in the model, and we have seen them occur with steep slopes in bed topography.  Examining the effects on creep closure is a wonderful suggestion that will be considered in upcoming work that focuses on real topography where negative pressures arise.  In the present paper, our intention is to simply lay out the model formulation with simple simulations (in which negative pressures do not occur).

7.12. How is the Pw=Pi restriction implemented without violating conservation of mass? If it's not, it would be good to report the amount of mass-conservation viola- tion this restriction imposes.

The pressure cap is imposed within  the nonlinear Picard iteration loop, so the system will iterate further by solving for the head field again with this cap activated at some computational nodes, and mass balance is thus always satisfied.

Note: the simulations included in this model development paper do not encounter pressures that run into the overburden limit.  In future tests on more complex terrain and with thicker ice, this should certainly be addressed in detail.

7.17. "Euler-Backward" => "backward Euler" seems more conventional, unless this means something else.

The text has been changed (page 9, line 23).

7.17. Picard iteration. This is a common methodology, but one not known for its speed. Though not mentioned in the manuscript, I surmise that a major advantage of this modeling approach (unified physics applied everywhere) over others could be its efficiency, but perhaps not, depending on the numerical implementation. This reader would be very interested to known if the model formulation had the potential to be fast, and whether the numerical implementation was designed with this in mind. Given the disparity in timescales between ice flow and water flow which typically necessitates comparatively small timesteps for hydrological models, many model users will be looking

for hydrological models that do not add unnecessary computational burden to their ice-sheet models.

The Picard iteration (or some form of nonlinear iteration) is a necessary component of an implicit backward Euler formulation. Picard iteration is much simpler to implement than a Newton iteration for this problem largely because of the complex nonlinearities for which Jacobian matrix computations for the latter will become very involved. There are limitations with Newton's method as well – non-convergence is often encountered.  For this reason, we implemented Picard iteration, but future computational enhancements of our model could consider alternative nonlinear solvers such as Jacobian-Free-Newton-Krylov methods.

We discuss the importance of time step in the new Section 4.1 of the revised manuscript ("Model Limitations").  Our model does require relatively small time steps compared to the years-scale of ice flow models.  At the same time, it should be noted that the time scales associated with important physical processes and forcing terms in ice sheet dynamics versus subglacial hydrology are indeed vastly different.  Attempting to run subglacial hydrology models on the same time-steps as used in ice sheet models will not be ideal, simply because the relevant physics are not properly resolved.  We anticipate that the future development of coupled ice-sheet and subglacial hydrology models will use different time-steps in each sub-model, and two-way coupling will not implemented in every subglacial hydrology model time-step.  Alternatively, seasonal time-scale coupled simulations could inform the representation of sliding in longer decadal and centurial simulations.

7.19-20. It sounds like gap height and hydraulic head are not solved simultaneously (or iteratively). Why not? Explicit time-stepping is simple but can lead to large errors. Can the authors reassure the readers that this has been investigated and propose corresponding limits on the time step?

We employ a semi-implicit solution strategy.  We solve for head with a nonlinear iterative approach in each time step, with the gap height from the end of the previous time-step held fixed.  Using the new head field, the gap height is updated before proceeding to the next time-step.  Thus, the gap height is being lagged by one time-step.  This is not an uncommon strategy in highly nonlinear problems, but it does place a restriction on the time-step in transient simulations (for steady input simulations, this is not an issue since the geometry and pressure converge to steady states).  Additional discussion of time-step size has been added in the revised manuscript (Section 4.1).

8.1. Is the model convergence sensitive to prescribed initial gap height?

No (at least for initial gap heights within reason).  For initial gap height ranging from at least 0.001 to 0.1 m, or randomly perturbed within that range, the model converges to the same state, and model run times vary only by a few seconds at most.  This has been added to Section 4.1 (Model Limitations).

8.7. Curious why closure is not included in "degree of channelization". If closure balances

opening at small gap heights for any opening mechanism, it seems channelization would be suppressed.

The degree of channelization is calculated in this way to be consistent with the way it was calculated for the Subglacial Hydrology Model Intercomparison Project (deFleurian et al., currently in review at *Journal of Glaciology*).  It was intended to give an idea of where the model behaves more as "sheetlike" vs. "channelized" based on the opening mechanism (to facilitate comparison to other models that differentiate between the two).

8.11-12. The software-style description seems a little strange. I guess the key thing here is that the output is ascii, not binary or something else? It seems like output from any model could be visualized in contour plots, timeseries, etc, and in any software.

Model output is binary, and the output could indeed be visualized in any software.  The text has been revised for clarity (page 10, lines 17-23).

8.15. Somewhere above this it should be noted that the mesh is irregular.

Text has been added earlier in Section 2.3 to state that the model can be run on a structured or unstructured mesh.  Using an unstructured mesh typically reduces bias in channel configuration, although the model performs well on either type.

8.16. "Application". Here I was expecting to see some multi-faceted demonstration of the model performance (e.g. accuracy, consistency, convergence, efficiency) prior to the demonstration that the model produces qualitatively familiar results in some basic tests. Model performance metrics are not often reported in journal articles focused on model applications, but I expected this would be different in Geoscientific Model Development. The Editor can decide if this suggestion is misguided; it could be misinterpreting the purpose and expectations of the journal.

We include results demonstrating convergence in the face of grid refinement (see Figs. 3, 4, 7, and 8) and time-step refinement (Fig. 10).  Due to the highly nonlinear nature of subglacial hydrology, it is not straightforward to compare the numerical simulations to any analytical solutions.  However, the model is part of a Subglacial Hydrology Model Intercomparison Project (SHMIP; deFleurian et al., under review), where it is being compared to other models on a range of benchmark problems.

If the editor has specific suggestions on a more formal evaluation of model performance beyond what we have included in the revised manuscript, we will be happy to consider them.

8-9. Subglacial hydrology models frequently have trouble in the presence of bed topography. The tests presented here omit bed topography with the exception of a gentle slope. It would be useful to know if this model does better than others in the presence of realistic bed topography. It's ok if it doesn't.

The model does perform well with bed topography, although negative water pressures are

calculated with very steep slopes (as mentioned in the text). The aim of this paper is to document the model formulation with simple illustrative example simulations. We have applied the model to simulate subglacial hydrology of the Store Glacier in west Greenland with real topography. That work that is beyond the scope of the current paper, but will be submitted soon.

9.3. "drainage configuration . . . affected by . . . bed topography" Except in this test the bed is flat. Is this just a general statement?

Yes, the text has been revised to reference how drainage configurations arise in general (page 11, lines 13-14).

9.4 "unstructured mesh" Please mention this when model implementation is described.

This has been added in Section 2.3 (see response to 8.15 above).

9.10. The test domains seem very small. It would be useful to report something on model runtimes. Is it practical to run this model coupled to an iceflow model for a large catchment?

The test domains are intentionally small to demonstrate features on a small scale. Model run times are dependent on the number of processors used and the type of machine, and will obviously increase with larger domains. Run time is mentioned in Section 3.1 of the revised manuscript, although we point out that run times will vary with model size, duration, complexity, time step size, number of processors, etc.

The SHAKTI model is best with relatively fine resolution to capture the effects of channelization, designed for simulation of individual glaciers, but it can be used on large domains. The new Section 4.1 includes some thoughts and guidance for selecting a mesh size.

10.3. "do not include storage term" Meaning englacial storage?

Correct. The simulations presented in this paper use zero englacial storage. This has been clarified in the text here (page 12, lines 18-19) and earlier (see response to 6.21).

10.6-7. Paper should make clear that ub is prescribed and constant, thus there is no two-way coupling with sliding, meaning the negative feedback associated with sliding is absent (Hoffman and Price, 2014?)

This has been clarified in the text (page 12, lines 23-27, also see Section 4.1).

10.9. This sounds like a problem that plagues most models (c.f. Downs et al, 2018: https://agupubs.onlinelibrary.wiley.com/doi/full/10.1002/2017JF004522), so should be noted as a common shortcoming with a citation or two.

We have mentioned the problem of low winter water pressures in subglacial hydrology

models at a few places in the revised text (page 4, lines 13-16; page 14, lines 15-17). Addressing this particular problem is not the aim of this model formulation and the illustrative examples included in this paper do not attempt to capture this behavior.

10.21/ "impose potential channel locations" Indeed this is a limitation in some models, but here the "channel" locations are a function of the mesh, just as they are in the models of Hewitt, Schoof and Werder. In the latter case, the channels may lie anywhere along the mesh edge. In this model, they may lie anywhere in the mesh elements. In the models of Hewitt, Schoof, Werder, increases in grid size mean small channels cannot be represented; in the current model, increases in grid size mean channels become unrealistically wide. Both are limitations in different ways.

It is true that gap height is calculated across elements, but we are not calculating a specific cross-sectional area of a channel, and a channel is not restricted to be along interface elements (see the intricate details in gap height distribution in Fig. 6). It is true that with a coarse grid, the apparent effects of channelized pathways become diffuse as shown in Figs. 3 and 6. We have included additional discussion in the revised manuscript of mesh dependence in Sections 3.2, 3.3, and 4.1.

10.25-26. This seems like a big deal, and a true potential advantage over the other models out there.

Yes, this is why we hope the model will be a useful contribution to the field. We thank the reviewer for recognizing the value in the inclusion of the dissipation term.

11.10-11. "Supports the notion" Too obvious. Suggest deleting.

Deleted.

11.13-14. Arguably the unconnected bed requires additional model physics, but this regime has been parameterized by Hoffman et al (2016) and Downs et al (2018).

We do not include specific physics to represent unconnected bed. However, in principle, the model formulation is capable of incorporating disconnected regions bounded by very small gap heights. A related issue is whether the numerical solution of the nonlinear Eq. (15) will be hampered by the occurrence of disconnected regions. We plan to investigate this issue further in future work based on strategies that we have employed previously in the context of rock fractures.

Additional citations have been added (page 14, line 17).

11.25 suggest "channels" => "pathways". Reword "sorts itself out".

The text has been revised.

2.22. "the model…, a model formulation" => "we describe the model formulation of SHaKTI, which allows for…"

Text revised.

5.19 and 5.25. These lines and the text that follows them do not form sentences.

Text revised.

Text revised.

References: more than 15 of the references are incomplete. Authors should check the list thoroughly. "Truffer" is missing an "r". "et al" is used where it probably shouldn't be. Sometimes journal titles are written out, sometimes they are not.

Thank you for alerting us to the fact that the reference list was not entirely accurate. We have updated the references.

Tables: check superscripts. I have great respect for SI, but please give ub in m/a also.

The tables have been updated.

[revised manuscript text omitted]

This form of the momentum equation can be derived based on laminar flow theory for flow between parallel plates. Integrating the Navier-Stokes equations twice to obtain the laminar flow for plane Poiseuille flow, we obtain:

$$q_{lam} = \frac{-b^3 g}{12\nu}\nabla h \tag{6}$$

where $\nu$ is the kinematic viscosity of water. The definition of Reynolds number follows the precedent in fracture literature, using the gap height $b$ as a characteristic length scale:

$$Re = \frac{|vb|}{\nu} = \frac{|\boldsymbol{q}|}{\nu} \tag{7}$$

where $v$ is the bulk velocity of the flow. The laminar flux (Eq. 6) is modified to allow for transition to a turbulent regime, and Eq. (5) can be written in the form:

$$\frac{\boldsymbol{q}}{\nabla h} = \frac{-b^3 g}{12\nu(1+\omega\frac{|\boldsymbol{q}|}{\nu})} = \frac{-b^3 g}{12(\nu+\omega|\boldsymbol{q}|)} \tag{8}$$

Using a value of $\omega = 0.001$, the transition to turbulent flow occurs around $Re = 1,000$. For turbulent flow with high Reynolds number ($Re >> 1,000$), $\omega Re >> 1$, and the flux $\boldsymbol{q}$ is proportional to the square root of the head gradient magnitude:

$$\boldsymbol{q}_{turb}^2 = \frac{-b^3 g}{12\omega}\nabla h \tag{9}$$

For laminar flow with low Reynolds number ($Re << 1,000$), $\omega Re << 1$, and $\boldsymbol{q}$ is proportional to the head gradient magnitude (Eq. 6). For intermediate Reynolds numbers in the wide transition between laminar and turbulent, $\boldsymbol{q}$ exhibits a combined proportionality to the head gradient or its square root.

Hydraulic head is calculated as:

[revised manuscript text omitted]

25  the time step increases, small stable fluctuations are seen. With dt=4 h, however, the model never converges to the solution, but instead enters a large stable oscillation between incorrect values. For larger time steps than dt=4 h, the nonlinear iteration itself has difficulty converging due to a blow-up in dissipation rates, which leads to an oscillation because of our cap on water pressure to not exceed ice overburden pressure. The appropriate time step size is dependent on particularities of a simulation such as topography, ice thickness, and meltwater input rates. Due to the highly nonlinear nature of the equations, it is unfortunately not

30  straightforward to establish a criterion for stable model behavior. As a general guideline we suggest conducting an initial test with a time step of 1 hour and adjusting accordingly. Implementing adaptive time stepping in SHAKTI could ease this process for users, but currently the adjustments are done manually. 
[revised manuscript text omitted]